# Protein language model-embedded geometric graphs power inter-protein contact prediction

**Yunda Si, Chengfei Yan***

School of Physics, Huazhong University of Science and Technology, Wuhan, China

**Abstract** Accurate prediction of contacting residue pairs between interacting proteins is very useful for structural characterization of protein–protein interactions. Although significant improvement has been made in inter-protein contact prediction recently, there is still a large room for improving the prediction accuracy. Here we present a new deep learning method referred to as PLMGraph-Inter for inter-protein contact prediction. Specifically, we employ rotationally and translationally invariant geometric graphs obtained from structures of interacting proteins to integrate multiple protein language models, which are successively transformed by graph encoders formed by geometric vector perceptrons and residual networks formed by dimensional hybrid residual blocks to predict inter-protein contacts. Extensive evaluation on multiple test sets illustrates that PLMGraph-Inter outperforms five top inter-protein contact prediction methods, including Deep-Homo, GLINTER, CDPred, DeepHomo2, and DRN-1D2D_Inter, by large margins. In addition, we also show that the prediction of PLMGraph-Inter can complement the result of AlphaFold-Multimer. Finally, we show leveraging the contacts predicted by PLMGraph-Inter as constraints for protein–protein docking can dramatically improve its performance for protein complex structure prediction.

**\*For correspondence:**
chengfeiyan@hust.edu.cn

**Competing interest:** The authors declare that no competing interests exist.

## eLife assessment

This study presents a **useful** deep learning-based inter-protein contact prediction method named PLMGraph-Inter which combines protein language models and geometric graphs. The evidence supporting the claims of the authors is **solid**. The authors show that their approach may be used in cases where AlphaFold-Multimer performs poorly. This work will be of interest to researchers working on protein complex structure prediction, particularly when accurate experimental structures are available for one or both of the monomers in isolation.

## Introduction

Protein–protein interactions (PPIs) are essential activities of most cellular processes (*Alberts, 1998*; *Spirin and Mirny, 2003*). Structure characterization of PPIs is important for mechanistic investigation of these cellular processes and therapeutic development (*Goodsell and Olson, 2000*). However, currently experimental structures of many important PPIs are still missing as experimental methods to resolve complex structures such as X-ray crystallography, nuclear magnetic resonance, and cryo-electron microscopy are costly and time-consuming (*Berman et al., 2000*). Therefore, it is necessary to develop computational methods to predict protein complex structures (*Bonvin, 2006*). Predicting contacting residue pairs between interacting proteins can be considered an intermediate step for protein complex structure prediction (*Hopf et al., 2014*; *Ovchinnikov et al., 2014*) as the predicted contacts can be integrated into protein–protein docking algorithms to assist protein complex structure

prediction (*Dominguez et al., 2003*; *Li and Huang, 2021*; *Sun et al., 2020*). Besides, the predicted contacts can also be very useful to guide protein interfacial design (*Martino et al., 2021*) and the inter-protein contact prediction methods can be further extended to predict novel PPIs (*Cong et al., 2019*; *Green et al., 2021*).

Based on the fact that contacting residue pairs often vary co-operatively during evolution, coevolutionary analysis methods *Weigt et al., 2009* have been used in previous studies to predict inter-protein contacts (*Hopf et al., 2014*; *Ovchinnikov et al., 2014*). However, coevolutionary analysis methods do have certain limitations. For example, effective coevolutionary analysis requires a large number of interolog sequences, which are often difficult to obtain, especially for heteromeric PPIs (*Hopf et al., 2014*; *Ovchinnikov et al., 2014*), and it is difficult to distinguish inter-protein and intra-protein coevolutionary signals for homomeric PPIs (*Uguzzoni et al., 2017*). Inspired by its great success in intra-protein contact prediction (*Hanson et al., 2018*; *Ju et al., 2021*; *Li et al., 2019*; *Si and Yan, 2021*; *Wang et al., 2017*), deep learning has also been applied to predict inter-protein contacts (*Guo et al., 2022*; *Roy et al., 2022*; *Xie and Xu, 2022*; *Yan and Huang, 2021*; *Zeng et al., 2018*). ComplexContact (*Zeng et al., 2018*), to the best of our knowledge, the first deep learning method for inter-protein contact prediction, has significantly improved the prediction accuracy over coevolutionary analysis methods. However, its performance on eukaryotic PPIs is still quite limited, partly due to the difficulty of accurately inferring interologs for eukaryotic PPIs. In a later study, coming from the same group as ComplexContact, Xie et al. developed GLINTER (*Xie and Xu, 2022*), another deep learning method for inter-protein contact prediction. Compared with ComplexContact, GLINTER leverages structures of interacting monomers, from which their rotational invariant graph representations are used as additional input features. GLINTER outperforms ComplexContact in the prediction accuracy, although there is still large room for improvement, especially for heteromeric PPIs. It is worth mentioning that CDPred (*Guo et al., 2022*), a recently developed method, further surpasses GLINTER in prediction accuracy with 2D attention-based neural networks. Apart from these methods developed to predict inter-protein contacts for both homomeric and heteromeric PPIs, inter-protein contact prediction methods specifically for homomeric PPIs were also developed (*Roy et al., 2022*; *Wu et al., 2022*; *Yan and Huang, 2021*) as predicting the inter-protein contacts for homomeric PPIs is generally much easier due to the symmetric restriction, relatively larger interfaces and the trivialness of inter-ologs identification. For example, Yan et al. developed DeepHomo (*Yan and Huang, 2021*), a deep learning method specifically to predict inter-protein contacts of homomeric PPIs, which also significantly outperforms coevolutionary analysis-based methods. However, DeepHomo requires docking maps calculated from structures of interacting monomers, which is computationally expensive and is also sensitive to the quality of monomeric structures. Besides, coming from the same group, *Lin et al., 2023* further developed DeepHomo2 for inter-protein contact prediction for homomeric PPIs by including the multiple sequence alignment (MSA) embeddings and attentions from an MSA-based protein language model (PLM) (MSA transformer) (*Rao et al., 2021b*) in their prediction model, which further improved the prediction performance. At almost the same time as DeepHomo2, we proved that embeddings from PLMs (*Rao et al., 2021a*; *Rives et al., 2021*) are very effective features in predicting inter-protein contacts for both homomeric and heteromeric PPIs, and we further show the sequence embeddings (ESM-1b [*Rives et al., 2021*]), MSA embeddings (ESM-MSA-1b [*Rao et al., 2021b*] and Position-Specific Scoring Matrix [PSSM]), and the inter-protein coevolutionary information complement each other in the prediction, with which we developed DRN-1D2D_Inter (*Si and Yan, 2023*). Extensive benchmark results show that DRN-1D2D_Inter significantly outperforms DeepHomo and GLINTER in inter-protein contact prediction, although DRN-1D2D_Inter makes the prediction purely from sequences.

In this study, we developed a structure-informed method to predict inter-protein contacts. Given the structures of two interacting proteins, we first build rotationally and translationally (SE(3)) invariant geometric graphs from the two monomeric structures, which encode both the inter-residue distance and orientation information of the monomeric structures. We further embedded the single-sequence embeddings (ESM-1b), MSA embeddings (ESM-MSA-1b and PSSM), and structure embeddings (ESM-IF [*Hsu et al., 2022*]) from PLMs in the graph nodes of the corresponding residues to build the PLM-embedded geometric graphs, which are then transformed by graph encoders formed by geometric vector perceptrons (GVPs) to generate graph embeddings for interacting monomers. The graph embeddings are further combined with inter-protein pairwise features and transformed by

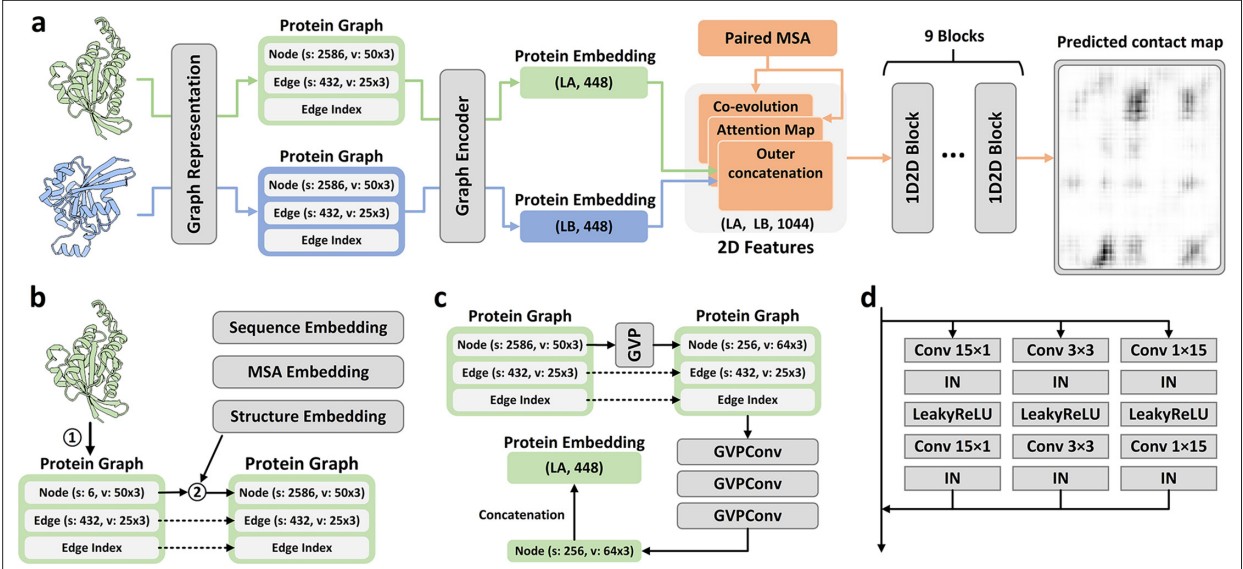

**Figure 1.** Overview of PLMGraph-Inter. (**a**) The network architecture of PLMGraph-Inter. (**b**) The graph representation module. (**c**) The graph encoder module, s denotes scalar features, v denotes vector features. (**d**) The dimensional hybrid residual block ('IN' denotes Instance Normalization).

The online version of this article includes the following figure supplement(s) for figure 1:

**Figure supplement 1.** The graph representation of protein structures.

residual networks formed by dimensional hybrid residual blocks (residual block hybridizing 1D and 2D convolutions) to predict inter-protein contacts. The developed method referred to as PLMGraph-Inter was extensively benchmarked on multiple tests with application of either experimental or predicted structures of interacting monomers as the input. The result shows that in both cases PLMGraph-Inter outperforms other top prediction methods, including DeepHomo, GLINTER, CDPred, Deep-Homo2, and DRN-1D2D_Inter by large margins. In addition, we also compared the prediction results of PLMGraph-Inter with the protein complex structures generated by AlphaFold-Multimer (*Evans et al., 2022*). The result shows that for many targets that AlphaFold-Multimer made poor predictions, PLMGraph-Inter yielded better results. Finally, we show leveraging the contacts predicted by PLMGraph-Inter as constraints for protein–protein docking can dramatically improve its performance for protein complex structure prediction.

## Results
### Overview of PLMGraph-Inter

The method of PLMGraph-Inter is summarized in *Figure 1a*. PLMGraph-Inter consists of three modules: the graph representation module (*Figure 1b*), the graph encoder module (*Figure 1c*), and the residual network module (*Figure 1d*). Each interacting monomer is first transformed into a PLM-embedded graph by the graph representation module, then the graph is passed through the graph encoder module to obtain a 1D representation of each protein. The two protein representations are transformed into 2D pairwise features through outer concatenation (horizontal and vertical tiling followed by concatenation) and further concatenated with other 2D pairwise features including the inter-protein attention maps and the inter-protein coevolution matrices, which are then transformed by the residual network module to obtain the predicted inter-protein contact map.

### The graph representation module

The first step of the graph representation module is to represent the protein 3D structure as a geometric graph, where each residue is represented as a node, and an edge is defined if the $C_\alpha$ atom distance between two residues is less than 18 Å. For each node and edge, we use scalars and vectors extracted from the 3D structures as their geometric features. To make the geometric graph SE(3) invariant, we use a set of local coordinate systems to extract the geometric vectors. The SE(3)

invariance of representation of each interacting monomer is important as in principle the inter-protein contact prediction result should not depend on the initial positions and orientations of protein structures. A detailed description can be found in the 'Methods' section. The second step is to integrate the single-sequence embedding from ESM-1b (*Rives et al., 2021*), the MSA embedding from ESM-MSA-1b (*Rao et al., 2021a*), the PSSM calculated from the MSA, and structure embedding from ESM-IF (*Hsu et al., 2022*) for each interacting monomer using its corresponding geometric graph. Where ESM-1b and ESM-MSA-1b are pretrained PLMs learned from large datasets of sequences and MSAs respectively with masked language modeling tasks, and ESM-IF is a supervised PLM trained from 12 million protein structures predicted by AlphaFold2 (*Jumper et al., 2021*) for fixed backbone design. The embeddings from these models contain high-dimensional representations of each residue in the protein, which are concatenated and further combined with the PSSM to form additional features of each node in the geometric graph. Since the sequence embeddings, MSA embeddings, PSSM, and structure embeddings are all SE(3) invariant, the PLM-embedded geometric graph of each protein is also SE(3) invariant.

## The graph encoder module

The graph encoder module is formed by GVP and GVP convolutional layer (GVPConv) (*Jing et al., 2021a*; *Jing et al., 2021b*). Where GVP is a graph neural network module consisting of a scalar track and a vector track, which can perform rotationally invariant transformations on scalar features and rotationally equivariant on vector features of nodes and edges; GVPConv follows the message passing paradigm of graph neural network and mainly consists of GVP, which updates the embedding of each node by passing information from its neighboring nodes and edges. A detailed description of GVP and GVPConv can be found in the 'Methods' section and also in the work of GVP (*Jing et al., 2021a*; *Jing et al., 2021b*). For each protein graph, we first use a GVP module to reduce the dimension of the scalar features of each node from 2586 to 256, which is then transformed successively by three GVPConv layers. Finally, we stitch the scalar features and the vector features of each node to form the 1D representation of the protein. Since the input protein graph is SE(3) invariant and the GVP and GVPConv transformations are rotationally equivariant, the 1D representation of each interacting monomer is also SE(3) invariant.

## The residual network module

The residual network module is mainly formed by nine-dimensional hybrid residual blocks to transform the 2D feature maps to obtain the predicted inter-protein contact map. Our previous study illustrated the effective receptive field can be enlarged with the application of the dimensional hybrid residual block, thus helping in improving the model performance (*Si and Yan, 2021*). A more detailed description of the transforming procedure can be found in the 'Methods' section.

## Evaluation of PLMGraph-Inter on HomoPDB and HeteroPDB test sets

We first evaluated PLMGraph-Inter on two self-built test sets which are non-redundant to the training dataset of PLMGraph-Inter: HomoPDB and HeteroPDB. Where HomoPDB is the test set for homomeric PPIs containing 400 homodimers and HeteroPDB is the test set for heteromeric PPIs containing 200 heterodimers. For comparison, we also evaluated DeepHomo, GLINTER, DeepHomo2, CDPred, and DRN-1D2D_Inter on the same datasets. Since DeepHomo and DeepHomo2 were developed to predict inter-protein contacts only for homomeric PPIs, their evaluation was only performed on HomoPDB. It should be noted that since HomoPDB and HeteroPDB are not de-redundant with the training sets of DeepHomo, DeepHomo2, CDPred, and GLINTER, the performances of the four methods may be overestimated.

In all the evaluations, the structural-related features were drawn from experimental structures of interacting monomers separated from complex structures of PPIs after randomizing their initial positions and orientations (DRN-1D2D_Inter does not use structural information). Besides, we also used the AlphaFold2 predicted monomeric structures as the input, considering experimental structures of interacting monomers often do not exist. Since the interacting monomers in HomoPDB and HeteroPDB are not de-redundant with the training set of AlphaFold2, using default settings of AlphaFold2 may overestimate its performance. To mimic the performance of AlphaFold2 in real practice and produce predicted monomeric structures with more diverse qualities, we only used the MSA searched

**Table 1.** The performances of DeepHomo, GLINTER, DRN-1D2D_Inter, DeepHomo2, CDPred, and PLMGraph-Inter on the HomoPDB and HeteroPDB test sets using experimental structures (AlphaFold2 predicted structures).

| Methods | HomoPDB (precision %) | | | | | HeteroPDB (precision %) | | | | |
|---|---|---|---|---|---|---|---|---|---|---|
| | L/5 | L/10 | 50 | 10 | 5 | L/5 | L/10 | 50 | 10 | 5 |
| DeepHomo | 43.2 (39.3) | 46.7 (42.7) | 42.4 (38.8) | 48.5 (44.8) | 49.9 (46.2) | | | | | |
| GLINTER | 42.9 (47.3) | 45.0 (50.1) | 42.2 (52.1) | 46.4 (51.9) | 48.5 (53.6) | 23.9 (25.1) | 24.7 (27.0) | 20.9 (21.9) | 25.5 (25.8) | 26.7 (26.2) |
| DRN-1D2D_Inter | 52.5 | 55.2 | 51.3 | 56.6 | 57.6 | 34.9 | 37.1 | 32.6 | 38.1 | 38.5 |
| DeepHomo2 | 55.6 (52.4) | 58.1 (53.9) | 55.0 (51.7) | 59.4 (55.7) | 61.3 (56.7) | | | | | |
| CDPred | 59.4 (54.7) | 61.3 (56.2) | 58.4 (54.1) | 62.4 (57.1) | 62.9 (57.7) | 30.0 (30.2) | 31.0 (31.7) | 27.6 (27.3) | 32.0 (32.2) | 32.1 (32.7) |
| PLMGraph-Inter | **68.6** (**61.8**) | **70.4** (**63.6**) | **67.3** (**60.9**) | **71.6** (**65.0**) | **72.1** (**65.25**) | **45.9** (**41.9**) | **48.6** (**43.6**) | **41.4** (**37.8**) | **49.1** (**44.1**) | **51.6** (**45.0**) |

The highest mean precision (%) in each column is highlighted in bold.

from Uniref100 ( *Suzek et al., 2015*) protein sequence database as the input of AlphaFold2 and set to not use the template. The predicted structures yielded a mean TM-score of 0.88, which is close to the performance of AlphaFold2 for CASP14 targets (mean TM-score 0.85) (*Lin et al., 2023*).

*Table 1* shows the mean precision of each method on the HomoPDB and HeteroPDB when the top (5, 10, 50, L/10, L/5) predicted inter-protein contacts are considered, where L denotes the sequence length of the shorter protein in the PPI. (Note that GLINTER encountered errors for 81 [5 when using the predicted monomeric structures] targets in HomoPDB and 15 [3 when using the predicted monomeric structures] targets in HeteroPDB at run time and did not produce predictions, thus we removed these targets in the evaluation of the performance of GLINTER. The performances of HomoPDB and HeteroPDB for these methods after the removal of these targets which GLINTER failed in any case are shown in *Supplementary file 1*.) As can be seen from the table, whenever the experimental or the predicted monomeric structures were used as the input, the mean precision of PLMGraph-Inter far exceeds those of other algorithms in each metric for both datasets. Particularly, the mean precision of PLMGraph-Inter is substantially improved in each metric on each dataset compared to our previous method DRN-1D2D_Inter, which used most features of PLMGraph-Inter except those drawn from the structures of the interacting monomers, illustrating the importance of the inclusion of structural information. Besides, GLINTER, CDPred, and DeepHomo2 also use structural information and PLMs, but have much lower performance than PLMGraph-Inter, illustrating the efficacy of our deep learning framework.

It can also be seen from the result that all the methods tend to have better performances on HomoPDB than those on HeteroPDB. One possible reason is that the complex structures of homodimers are generally C2 symmetric, which largely restricts the configurational spaces of PPIs, making the inter-protein contact prediction a relatively easier task (e.g., the inter-protein contact maps for homodimers are also symmetric). Besides, compared with heteromeric PPIs, we may be more likely to successfully infer the inter-protein coevolutionary information for homomeric PPIs to assist the contact prediction for two reasons: first, it is straightforward to pair sequences in the MSAs for homomeric PPIs, thus the paired MSA for homomeric PPIs may have higher qualities; second, homomeric PPIs may undergo stronger evolutionary constraints, as homomeric PPIs are generally permanent interactions, but many heteromeric PPIs are transient interactions.

In addition to using the mean precision on each test set to evaluate the performance of each method, the performance comparisons between PLMGraph-Inter and other models on the top 50 predicted contacts for each individual target in HomoPDB and HeteroPDB are shown in *Figure 2* (separate comparisons are shown in *Figure 2—figure supplements 1* and *2*). Specifically, when the experimental (predicted) structures were used as the input, PLMGraph-Inter achieved the best performance for 60% (53%) of the targets in HomoPDB and 58% (51%) of the targets in HeteroPDB.

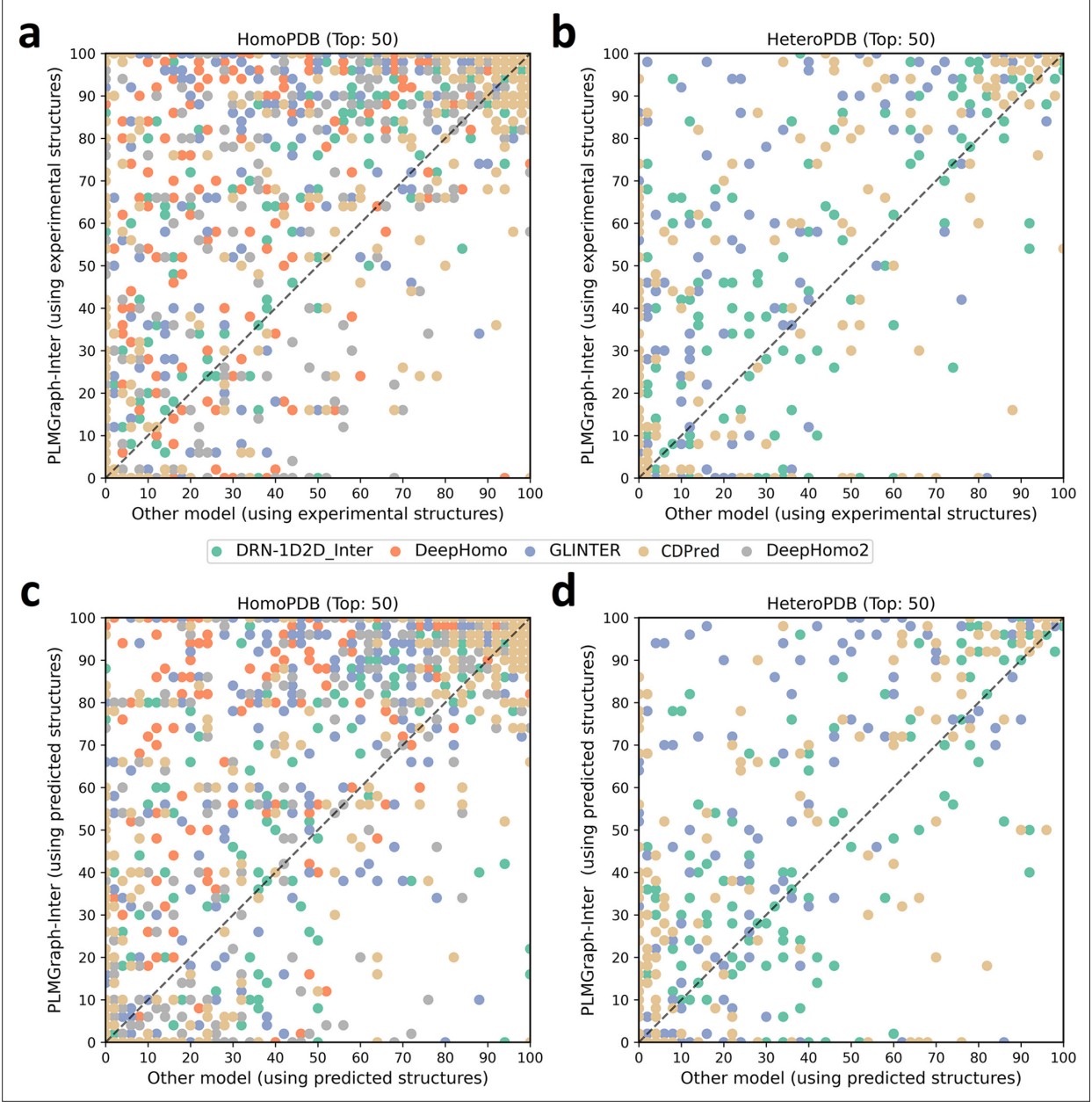

**Figure 2.** The performances of PLMGraph-Inter and other methods on the HomoPDB and HeteroPDB test sets. (**a, b**) The head-to-head comparison of the precisions (%) of the top 50 contacts predicted by PLMGraph-Inter and other methods for each target in (**a**) HomoPDB and (**b**) HeteroPDB using experimental structures. (**c, d**) The head-to-head comparison of the precisions (%) of the top 50 contacts predicted by PLMGraph-Inter and other methods for each target in (**c**) HomoPDB and (**d**) HeteroPDB using AlphaFold2 predicted structures.

The online version of this article includes the following figure supplement(s) for figure 2:

**Figure supplement 1.** The head-to-head comparison of the precisions (%) of the top 50 contacts predicted by PLMGraph-Inter and other methods (**a**: DRN-1D2D_Inter; **b**: DeepHomo; **c**: GLINTER ; **d**: CDPred; **e**: DeepHomo2) for each target in HomoPDB and HeteroPDB using experimental structures.

**Figure supplement 2.** The head-to-head comparison of the precisions (%) of the top 50 contacts predicted by PLMGraph-Inter and other methods (**a**: DRN-1D2D_Inter; **b**: DeepHomo; **c**: GLINTER ; **d**: CDPred; **e**: DeepHomo2) for each target in HomoPDB and HeteroPDB using AlphaFold2 predicted structures.

**Figure supplement 3.** The mean precision versus contact density for the top 50 contacts predicted by PLMGraph-Inter, GLINTER, DeepHomo, DeepHomo2, CDPred, and DRN-1D2D_Inter on the HomoPDB test set (**a, c**) and HeteroPDB test set (**b, d**) using experimental structures (first row) and AlphaFold2 predicted structures (second row).

*Figure 2 continued on next page*

*Figure 2 continued*

**Figure supplement 4.** The mean precision versus log ($N_{eff}^{norm}$) for the top 50 contacts predicted by PLMGraph-Inter, GLINTER, DeepHomo, DeepHomo2, and CDPred, DRN-1D2D_Inter on the HomoPDB test set (**a, c**) and HeteroPDB test set (**b, d**) using experimental structures (first row) and

*Figure 2 continued on next page*

*Figure 2 continued*

AlphaFold2 predicted structures (second row).

We further group targets in each dataset according to their inter-protein contact densities defined as $\frac{\#\ of\ contacts}{L_A * L_B}$ and the normalized number of the effective sequences ($N_{eff}^{norm}$) of paired MSAs. We found that all the methods tend to have lower performances for targets with lower contact densities (*Figure 2—figure supplement 3*), which is reasonable since obviously it is more challenging to identify the true contacts when their ratio is lower. We also found when the $N_{eff}^{norm}$ is low (log ($N_{eff}^{norm}$) < 3), the prediction performances of all methods tend to improve with $N_{eff}^{norm}$, but when $N_{eff}^{norm}$ reaches certain thresholds (log ($N_{eff}^{norm}$) > 4), the performances of all the methods tend to fluctuate with $N_{eff}^{norm}$

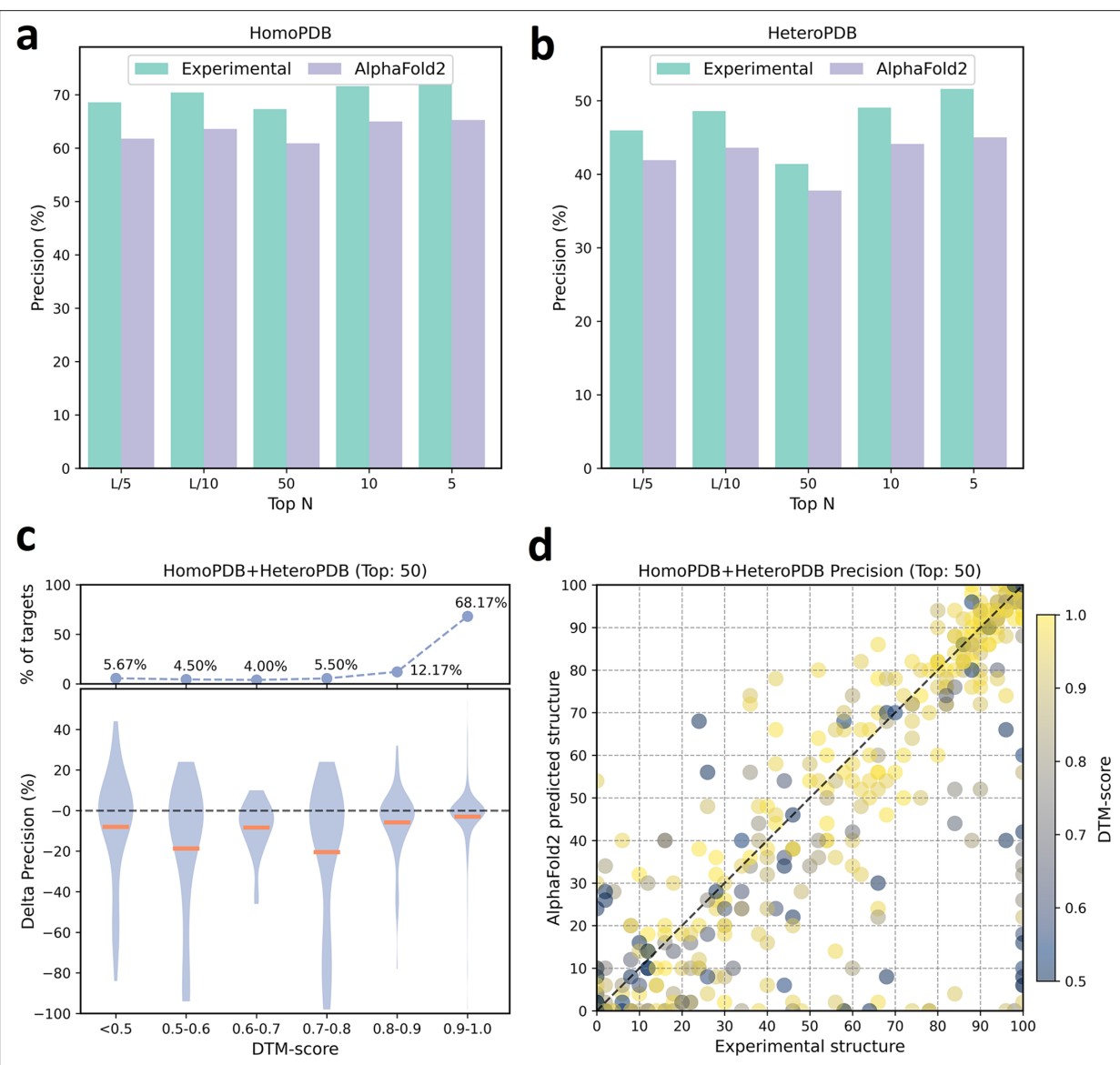

**Figure 3.** The performances of PLMGraph-Inter when using experimental and AlphaFold2 predicted structures as the input. (**a, b**) The performance comparison of PLMGraph-Inter when using experimental structures and AlphaFold2 predicted structures as the input on (**a**) HomoPDB and (**b**) HeteroPDB. (**c**) The performance gaps (measured as the difference of the mean precision of the top 50 predicted contacts) of PLMGraph-Inter with the application of AlphaFold2 predicted structures and experimental structures as the input when the protein–protein interactions (PPIs) are within different intervals of DTM-score. The upper panel shows the percentage of the total number of PPIs in each interval. (**d**) The comparison of the precision of the top 50 contacts predicted by PLMGraph-Inter for each target when using experimental structures and AlphaFold2 predicted structures as the input.

(*Figure 2—figure supplement 4*). However, PLMGraph-Inter consistently achieved the best performances in most categories.

## Impact of the monomeric structure quality on contact prediction

We further analyzed the performance difference of PLMGraph-Inter when using the AlphaFold2 predicted structures and using the experimental structures as the inputs. As shown in *Figure 3a and b*, when the predicted structures were used by PLMGraph-Inter for inter-protein contact prediction, mean precisions of the predicted inter-protein contacts in each metric on both HomoPDB and HeteroPDB test sets decreased by about 5% (also see *Table 1*), indicating qualities of the input structures do have a certain impact on the prediction performance.

We further explored the impact of the monomeric structure quality on the inter-protein contact prediction performance of PLMGraph-Inter. Specifically, for a given PPI, the TM-score (*Zhang and Skolnick, 2004*) was used to evaluate the quality of the predicted structure for each interacting monomer, and the TM-score of the predicted structure with lower quality was used to evaluate the overall monomeric structure prediction quality for the PPI, denoted as 'DTM-score'. In *Figure 3c*, we show the performance gaps (using the mean precisions of the top 50 predicted contacts as the metric) between applying the predicted structures and applying the experimental structures in the inter-protein contact prediction, in which we grouped targets according to DTM-scores of their monomeric structure prediction, and in *Figure 3d*, we show the performance comparison for each specific target. From *Figure 3c and d*, we can clearly see that when the DTM-score is lower, the prediction using the prediction structure tends to have lower accuracy. However, when the DTM-score is ≥0.8, there is almost no difference between applying the predicted structures and applying the experimental structure, which shows the robustness of PLMGraph-Inter to the structure quality.

## Ablation study

To explore the contribution of each input component to the performance of PLMGraph-Inter, we conducted an ablation study on PLMGraph-Inter. The graph representation from the structure of each interacting proteins is the base feature of PLMGraph-Inter, so we first trained the baseline model using only the geometric graphs as the input feature, denoted as model a. Our previous study in DRN-1D2D_Inter has shown that the single-sequence embeddings, MSA 1D features (including the MSA embeddings and PSSMs), and 2D pairwise features from the paired MSA play important roles in the model performance. To further explore the importance of these features when integrated with the geometric graphs, we trained models b–d separately (model b: geometric graphs + sequence embeddings; model c: geometric graphs + sequence embeddings + MSA 1D features; model d: geometric graphs + sequence embeddings + MSA 1D features + 2D features). Finally, we included the structure embeddings as additional features to train the model e (model e uses all the input features of PLMGraph-Inter). All the five models were trained using the same protocol as PLMGraph-Inter on the same training and validation partition without cross-validation. We further evaluated performances of models a–e together with PLMGraph-Inter (i.e., model f: model e + cross-validation) on HomoPDB and HeteroPDB using experimental structures of interacting monomers respectively.

In *Figure 4a*, we show the mean precisions of the top 50 predicted contacts by models a–f on HomoPDB and HeteroPDB, respectively. It can be seen from *Figure 4a* that including the sequence embeddings in the geometric graphs has a very good boost to the model performance (model b versus model a), while the additional introduction of MSA 1D features and 2D pairwise features can further improve the model performance (model d versus model c versus model b). DRN-1D2D_Inter also uses the same set of sequence embeddings, MSA 1D features, and 2D pairwise features as the input, and our model d shows a significant performance improvement over DRN-1D2D_Inter (single model) (the model trained on the same training and validation partition without cross-validation) on both HomoPDB and HeteroPDB (the mean precision improvement: HomoPDB: 14%; HeteroPDB: 5.6%), indicating that the introduced graph representation is important for the model performance. The head-to-head comparison of model d and DRN-1D2D_Inter (single model) on each specific target in *Figure 4b* further demonstrates the value of the graph representation. Besides, the additionally introduced structure embeddings from ESM-IF can further improve the mean precisions of the predicted contacts by 3–4% on both HomoPDB and HeteroPDB (model e versus model d) and the

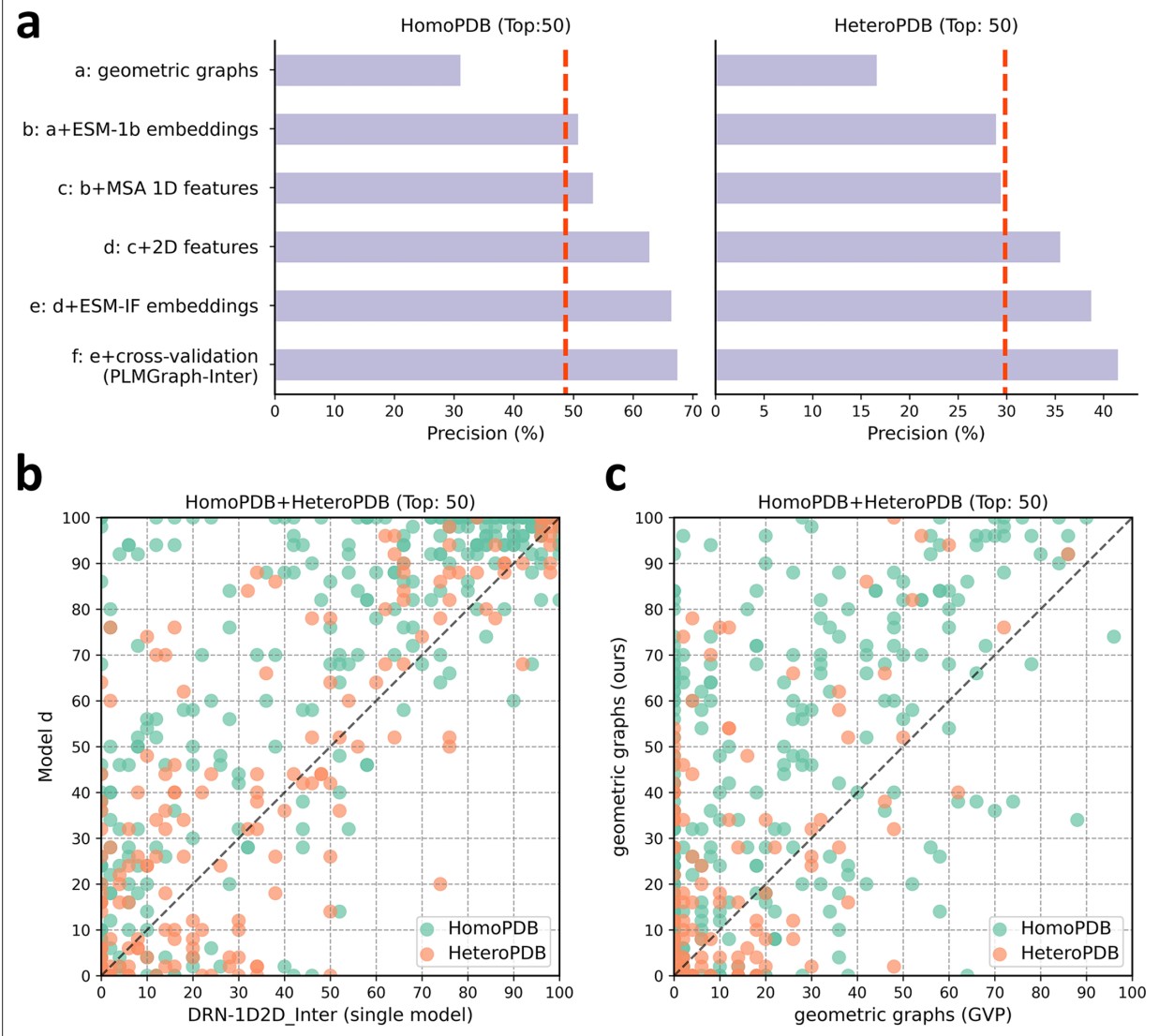

**Figure 4.** The ablation study of PLMGraph-Inter on the HomoPDB and HeteroPDB test sets. (**a**) The mean precisions of the top 50 contacts predicted by different ablation models on the HomoPDB and HeteroPDB test sets. (**b**) The head-to-head comparisons of mean precisions of the top 50 contacts predicted by model d and DRN-1D2D_Inter (single model) for each target in HomoPDB and HeteroPDB. (**c**) The head-to-head comparison of mean precisions of the top 50 contacts predicted by the model using our geometric graphs and the geometric vector perceptron (GVP) geometric graphs.

application of the cross-validation can also improve the precisions by 1.0% on HomoPDB and 2.7% on HeteroPDB (model f versus model d) (see *Supplementary file 2*).

To demonstrate the efficacy of our proposed graph representation of protein structures, we also trained a model using the structural representation proposed in the work of GVP (*Jing et al., 2021a*; *Jing et al., 2021b*) (denoted as 'GVP Graph'), as a control. Our structural representation differs significantly from GVP Graph. For example, we extracted inter-residue distances and orientations between five atoms (C,O,$C\alpha$,N, and a virtual $C\beta$) from the structure as the geometric scalar and vector features, in which the vector features are calculated in a local coordinate system. However, GVP Graph only uses the distances and orientations between $C$ atoms as the geometric scalar and vector features and the vector features are calculated in a global coordinate system. In addition, after the geometric graph is transformed by the graph encoder module, GVP Graph only uses the scalar features of each node as the node representation, while we concatenate the scalar and vector features of the node as the node representation. In *Figure 4c*, we show the performance comparison between this model and our base model (model a). From *Figure 4c*, we can clearly see that our base model significantly outperforms

**Table 2.** The performance of PLMGraph-Inter when using different sequence identity and fold similarity thresholds to further remove potential redundancies in HomoPDB and HeteroPDB.

| | | HomoPDB | | HeteroPDB | |
|---|---|---|---|---|---|
| | | Count | Precision (Top 50 [%]) | Count | Precision (Top 50 [%]) |
| | Original | 400 | 67.3 (60.9) | 200 | 41.4 (37.8) |
| | 40% | 341 | 68.7 (62.5) | 160 | 38.6 (35.6) |
| | 30% | 257 | 64.7 (58.7) | 144 | 38.1 (35.1) |
| Sequence identity (MMSeqs2) | 20% | 211 | 63.2 (56.3) | 138 | 38.5 (35.3) |
| | 10% | 211 | 63.2 (56.3) | 138 | 38.5 (35.3) |
| | 0.9 | 370 | 65.2 (58.3) | 185 | 39.7 (35.7) |
| | 0.8 | 281 | 61.8 (53.4) | 153 | 38.1 (34.1) |
| | 0.7 | 179 | 56.5 (45.8) | 126 | 38.8 (34.6) |
| Fold similarity (TM-align) | 0.6 | 124 | 50.4 (39.9) | 102 | 37.4 (34.1) |
| | 0.5 | 70 | 49.6 (41.3) | 83 | 36.5 (34.5) |

The results using experimental structures are shown outside the parentheses, and the results using the AlphaFold2 predicted structures are shown inside the parentheses.

the GVP Graph-based model on both HomoPDB and HeteroPDB, illustrating the high efficacy of our proposed graph representation.

We also explored the performance of PLMGraph-Inter on the HomoPDB and HeteroPDB test sets when using different protocols to further remove potential redundancies between the training and the test sets. Specifically, although the '40% sequence identity' used in our study is a widely used threshold to remove redundancy when evaluating deep learning-based PPI and protein complex structure prediction methods (*Evans et al., 2022*; *Sledzieski et al., 2021*), it is worth testing whether PLMGraph-Inter can keep its performance when more stringent threshold is applied. Besides, it is also worth evaluating whether PLMGraph-Inter can keep its performance on targets for which the folds of their interacting monomers are different from the targets in the training set (i.e., non-redundant in main chain structures of interacting monomers). To the best of our knowledge, all the previous studies failed to remove potential redundancies in folds of the interacting monomers when evaluating their methods.

In *Table 2*, we show mean precisions of the contacts (top 50) predicted by PLMGraph-Inter on the HomoPDB and HeteroPDB when various sequence identity thresholds (with MMseq2 [*Steinegger and Söding, 2018*]) and fold similarity thresholds (with TMalign [*Zhang and Skolnick, 2005*]) were further used in the de-redundancy (see section 'Further potential redundancies removal between the training and the test'). It can be seen from that table that when using more stringent sequence identity thresholds for de-redundancy, the performance of PLMGraph-Inter on both the HomoPDB and HeteroPDB datasets decreases very little. For example, even when using '10% sequence identity' for de-redundancy, mean precisions of the predicted contacts only decrease by 2–4%. Whereas when using fold similarities of the interacting monomers for de-redundancy, although the performance of PLMGraph-Inter on HeteroPDB decreases very little (only 3–4% when TM-score 0.5 is used as the threshold), the performance of PLMGraph-Inter on HomoPDB decreases significantly (17–19% when TM-score 0.5 is used as the threshold). One possible reason for the performance decrease on HomoPDB is that the binding mode of the homomeric PPI is largely determined by the fold of its monomer, thus the model does not generalize well on targets whose folds have never been seen during the training.

## Evaluation of PLMGraph-Inter on DHTest and DB5.5 test sets

We further evaluated PLMGraph-Inter on DHTest and DB5.5. The DHTest test set was formed by removing PPIs redundant to our training set from the original test set of DeepHomo, which contains 130 homomeric PPIs. The DB5.5 test set was formed by removing PPIs redundant to our training

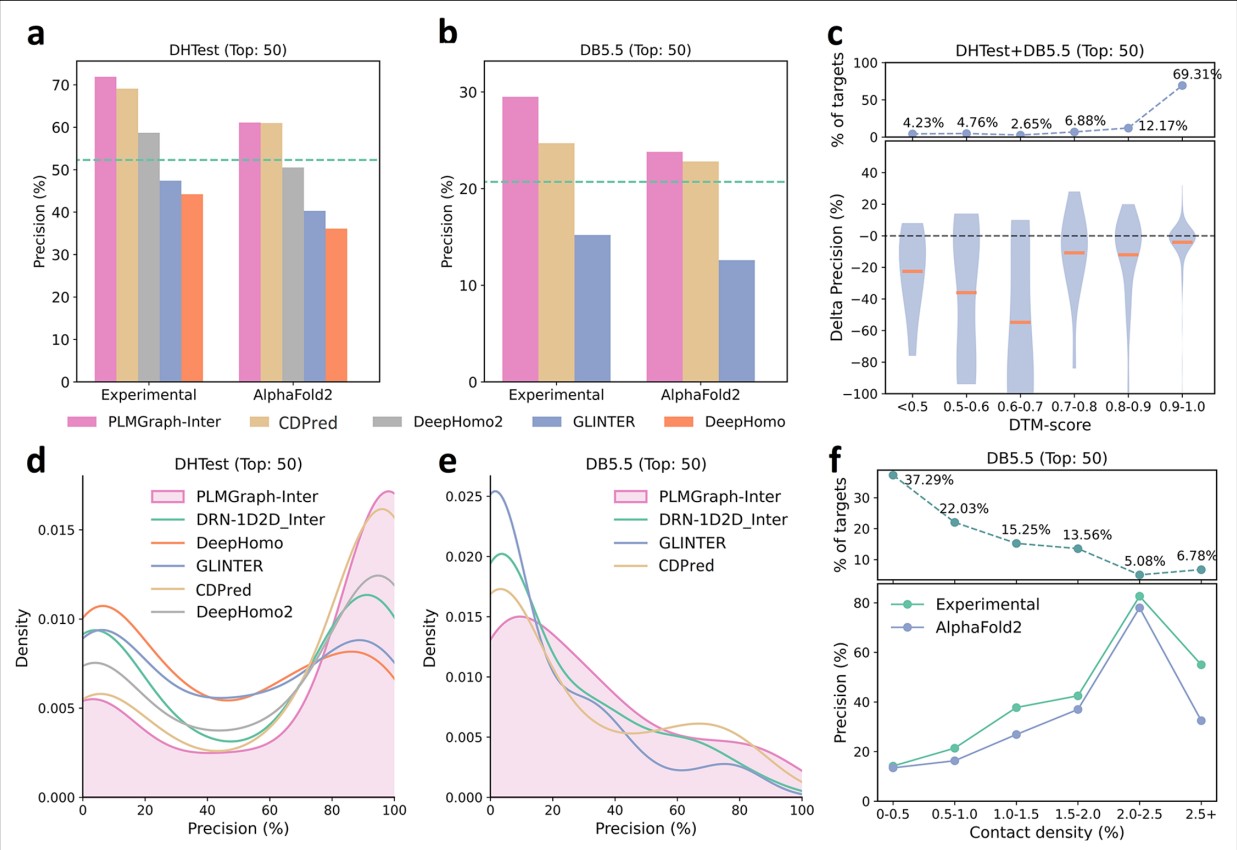

**Figure 5.** The performances of PLMGraph-Inter and other methods on the DHTest and DB5.5 test sets. (**a, b**) The mean precisions of the top 50 contacts predicted by PLMGraph-Inter, GLINTER, DeepHomo2, CDPred, and DeepHomo on (**a**) DHTest and (**b**) DB5.5 when using experimental structures and AlphaFold2 predicted structures as the input, where the green lines indicate the performance of DRN-1D2D_Inter. (**c**) The performance gaps (measured as the difference of the mean precision of the top 50 predicted contacts) of PLMGraph-Inter with the application of AlphaFold2 predicted structures and experimental structures as the input when the protein–protein interactions (PPIs) are within different intervals of DTM-score. The upper panel shows the percentage of the total number of PPI's in each interval. (**d, e**) The distributions of precisions of the top 50 contacts predicted by PLMGraph-Inter and other methods for PPIs in (**d**) DHTest and (**e**) DB5.5. (**f**) The mean precisions of the top 50 contacts predicted by PLMGraph-Inter on PPIs within different intervals of contact densities in DB5.5. The upper panel shows the percentage of the total number of PPIs in each interval.

dataset from the heterodimers in Protein-protein Docking Benchmark 5.5, which contains 59 hetero-meric PPIs. Still, both the experimental structures and the predicted structures (generated using the same protocol as in HomoPDB and HeteroPDB) of the interacting monomers were used respectively in the inter-protein contact prediction. It should be noted that since DHTest and DB5.5 are not de-redundant with the training sets of CDPred and GLINTER, particularly, all PPIs in the DHTest test set are included in the training set of CDPred, thus the performances of the two methods may be overestimated.

As shown in *Figure 5a and b*, when using the experimental structures in the prediction, the mean precisions of the top 50 contacts predicted by PLMGraph-Inter are 71.9% on DHTest and 29.5% on DB5.5 (also see *Supplementary file 3*), which are dramatically higher than DeepHomo, GLINTER, DeepHomo2, and DRN-1D2D_Inter. (Note that GLINTER encountered errors for 47 targets in DHTest and 3 targets in DB5.5 at run time and did not produce predictions, thus we removed these targets in the evaluation of the performance of GLINTER. The performances of other methods on DHTest and DB5.5 after the removal of these targets are shown in *Supplementary file 4*.) We can also see that although PLMGraph-Inter achieved significantly better performance than CDPred on DB5.5, its performance on DHTest is quite close to CDPred. However, it should be noted that the performance of CDPred on DHTest might be grossly overestimated since PPIs in DHTest are fully included in the training set of CDPred. The distributions of the precisions of the top 50 predicted contacts by

different methods on DHTest and DB5.5 are shown in *Figure 5d and e*, from which we can clearly see that PLMGraph-Inter can make high-quality predictions for more targets on both DHTest and DB5.5.

When using the predicted structures in the prediction, the mean precisions of the top 50 contacts predicted by PLMGraph-Inter show a reasonable decrease to 61.1% on DHTest and 23.8% on DB5.5, respectively (see *Figure 5a and b* and *Supplementary file 3*). We also analyzed the impact of the monomeric structure quality on the inter-protein contact prediction performance of PLMGraph-Inter. As shown in *Figure 5c*, when the DTM-score is ≥0.7, there is almost no difference between applying the predicted structures and the experimental structures, which is consistent with our analysis in HomoPDB and HeteroPDB.

We noticed that the performance of PLMGraph-Inter on the DB5.5 is significantly lower than that on HeteroPDB, and so are the performances of other methods. That the targets in DB5.5 have relatively lower mean contact densities (1.01% vs 1.29%) may partly explain this phenomenon. In *Figure 5f*, we show the variations of the precisions of predicted contacts with the variation of contact density. As can be seen from *Figure 5f*, as the contact density increases, precisions of predicted contacts tend to increase regardless of whether the experimental structures or predicted structures are used in the prediction. 37.29% targets in DB5.5 are with inter-protein contact densities lower than 0.5%, for which precisions of predicted contacts are generally very low, making the overall inter-protein contact prediction performance on DB5.5 relatively low.

## Comparison of PLMGraph-Inter with AlphaFold-Multimer

After the development of AlphaFold2, DeepMind also released AlphaFold-Multimer, as an extension of AlphaFold2 for protein complex structure prediction. The inter-protein contacts can also be extracted from the complex structures generated by AlphaFold-Multimer. It is worth making a comparison between the performances of AlphaFold-Multimer and PLMGraph-Inter on inter-protein contact prediction. Therefore, we also employed AlphaFold-Multimer (version 2.2) with its default settings to generate complex structures for all the PPIs in the four datasets which we used to evaluate PLMGraph-Inter. We then selected the 50 inter-protein residue pairs with the shortest heavy atom distances in each generated protein complex structures as the predicted inter-protein contacts. It should be noted that AlphaFold-Multimer used all protein complex structures in the Protein Data Bank deposited before April 30, 2018, in the model training, thus these PPIs may have a large overlap with the training set of AlphaFold-Multimer. Therefore, there is no doubt that the performance of AlphaFold-Multimer would be overestimated here. It should also be noted that although AlphaFold-Multimer makes the prediction from sequences, it automatically searches templates of the interacting monomers. When we checked our AlphaFold-Multimer runs, we noticed for 99% of the targets (including all the targets in the four datasets: HomoPDB, HeteroPDB, DHTest, and DB5.5), at least 20 templates were identified (AlphaFold-Multimer only employed the top 20 templates), and AlphaFold-Multimer employed the native template (i.e., the template which has the same PDB id with the target) for 87.8% of the targets. Besides, AlphaFold-Multimer employs multiple sequence databases including a huge metagenomics database (*Jumper et al., 2021*), but PLMGraph-Inter only employs the UniRef100, thus the comparison is not on the same footing.

In *Figure 6a*, we show the relationship between the quality of the generated protein complex structure (evaluated with DockQ) and the precision of the top 50 inter-protein contacts extracted from the protein complex structure predicted by AlphaFold-Multimer for each PPI in the homomeric PPI (DHTest + HomoPDB) and heteromeric PPI (DB5.5 + HeteroPDB) datasets. As can be seen from the figure, the precision of the predicted contacts is highly correlated with the quality of the generated structure. Especially when the precision of the contacts is higher than 50%, most of the generated complex structures have at least acceptable qualities (DockQ ≥ 0.23), in contrast, almost all the generated complex structures are incorrect (DockQ < 0.23) when the precision of the contacts is below 50%. Therefore, 50% can be considered as a critical precision threshold for inter-protein contact prediction (the top 50 contacts).

In *Figure 6b*, we show the comparison of the precisions of top 50 contacts predicted by AlphaFold-Multimer and PLMGraph-Inter for each target when using the experimental monomeric structures as the input for PLMGraph-Inter respectively (also see *Supplementary file 5*, and the comparison when using the AlphaFold2 predicted structures is shown in *Figure 6—figure supplement 1b*). It can be seen from the figure that although for most of the targets AlphaFold-Multimer yielded

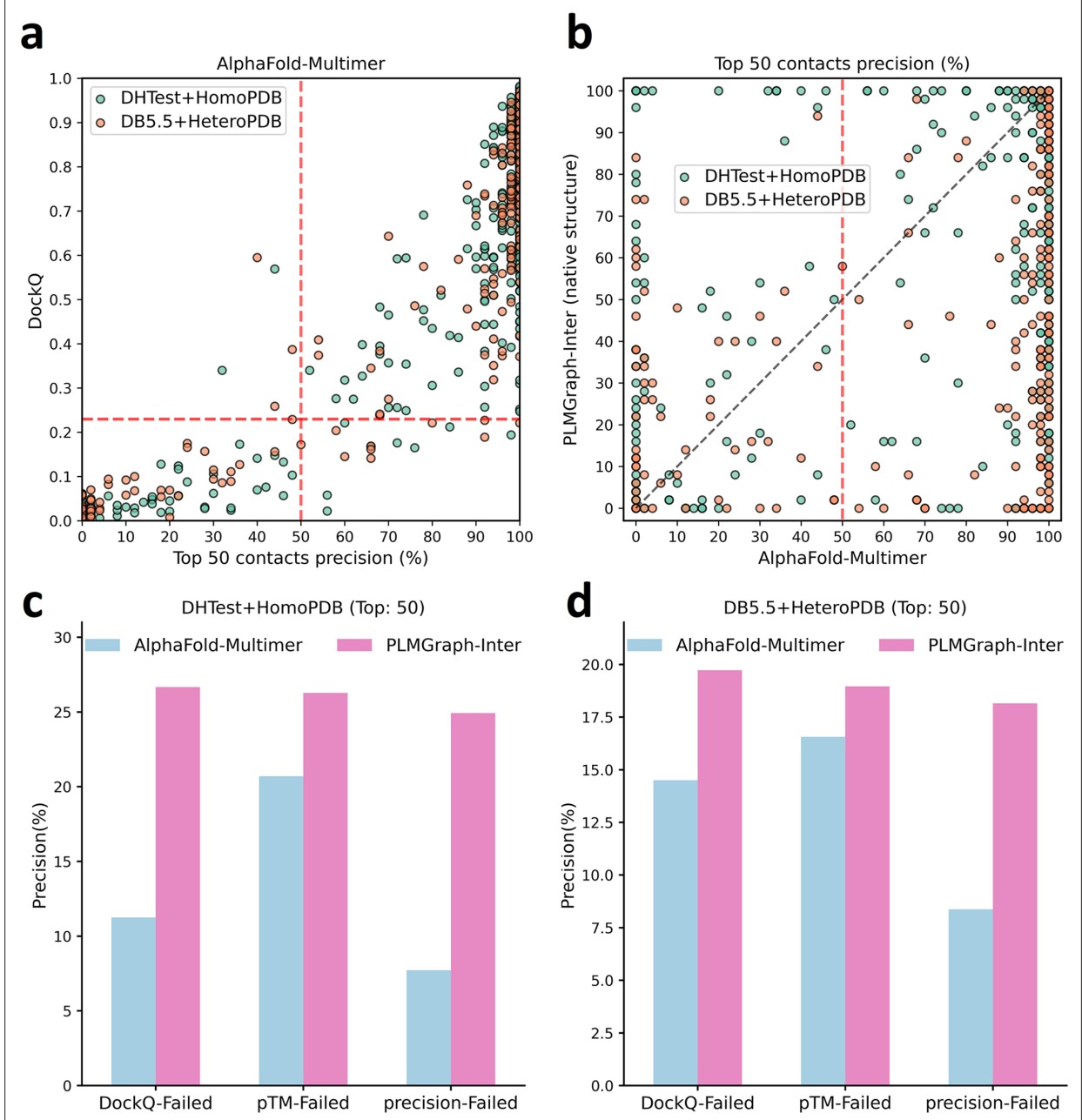

**Figure 6.** The comparison of PLMGraph-Inter with AlphaFold-Multimer. (**a**) The head-to-head comparison between the qualities of the protein complex structures generated by AlphaFold-Multimer (evaluated with DockQ) and the precision of the top 50 inter-protein contacts extracted from the generated protein complex structures. The red horizontal lines represent the threshold (DockQ = 0.23) to determine whether the complex structure prediction is successful or not. (**b**) The head-to-head comparisons of precisions of the top 50 inter-protein contacts predicted by PLMGraph-Inter and AlphaFold-Multimer for each target in the homomeric protein–protein interaction (PPI) and heteromeric PPI datasets. (**c, d**) The mean precisions of top 50 inter-protein contacts predicted by PLMGraph-Inter and AlphaFold-Multimer on the PPI subsets from (**c**) 'DHTest + HomoPDB' and (**d**) 'DB5.5 + HeteroPDB' in which the precision of the top 50 inter-protein contacts predicted by AlphaFold-Multimer is lower than 50% or the DockQ of the complex structure predicted by AlphaFold-Multimer is lower than 0.23 or the 'iptm + ptm' of the complex structure predicted by AlphaFold-Multimer is lower than 0.5.

The online version of this article includes the following figure supplement(s) for figure 6:

**Figure supplement 1.** The comparison of PLMGraph-Inter with AlphaFold-Multimer.

better results, but for a significant number of the targets AlphaFold-Multimer made poor predictions (precision <50%), the results of PLMGraph-Inter can have certain improvement over the AlphaFold-Multimer predictions.

We further explored the performance of PLMGraph-Inter on the PPIs which AlphaFold-Multimer failed to make correct predictions. Specifically, we denoted a PPI for which the precision of the top 50 inter-protein contacts predicted by AlphaFold-Multimer is lower than 50% or the DockQ of protein complex structure predicted by AlphaFold-Multimer is less than 0.23 as 'precision-Failed', 'DockQ-Failed'. The 'iptm + ptm' metrics output by AlphaFold-Multimer for each target also has a certain ability to characterize the quality of the predicted complexes. Our DockQ versus 'iptm + ptm' analysis shows that 0.5 can be reasonably chosen as the cutoff of 'iptm + ptm' to evaluate whether the prediction of AlphaFold-Multimer is successful or not (see *Figure 6—figure supplement 1a*), so we denoted a prediction for which the 'iptm + ptm' of the prediction is lower than 0.5 as 'pTM-Failed'. Then the mean precisions of top 50 contacts predicted by PLMGraph-Inter and AlphaFold-Multimer on the 'precision-Failed', 'pTM-Failed', and 'DockQ-Failed' sub-test sets from 'DHTest + HomoPDB' and 'DB5.5 + HeteroPDB' are shown in *Figure 6c and d* (see *Figure 6—figure supplement 1c and d* for the comparison when using the AlphaFold2 predicted structures). From *Figure 6c and d* we can see that the mean precisions of contacts predicted by PLMGraph-Inter are higher than the mean precisions of contacts predicted by AlphaFold-Multimer, further demonstrating that PLMGraph-Inter can complement AlphaFold-Multimer in certain cases.

## PLMGraph-Inter can significantly improve protein–protein docking performance

Prior to AlphaFold-Multimer, protein–protein docking is generally used for protein complex structure prediction. HADDOCK (*Honorato et al., 2021*; *van Zundert et al., 2016*) is a widely used information-driven protein–protein docking approach to model complex structures of PPIs, which allows us to encode predicted inter-protein contacts as constraints to drive the docking. In this study, we used HADDOCK (version 2.4) to explore the contribution of PLMGraph-Inter to protein complex structure prediction.

We prepared the test set of homomeric PPIs by merging HomoPDB and DHTest and the test set of heteromeric PPIs by merging HeteroPDB and DB5.5, where the monomeric structures generated previously by AlphaFold2 were used as the input to HADDOCK for protein–protein docking, in which the top 50 contacts predicted by PLMGraph-Inter with the application of the predicted monomeric structures were used as the constraints. Since HADDOCK generally cannot model large conformational changes in protein–protein docking, we filtered PPIs in which either of the AlphaFold2-generated interacting monomeric structure has a TM-score lower than 0.8. Finally, the homomeric PPI test set contains 462 targets, denoted as homodimer, and the heteromeric PPI test set contains 174 targets, denoted as heterodimer.

For each PPI, we used the top 50 contacts predicted by PLMGraph-Inter and other methods as ambiguous distance restraints between the alpha carbons $(C\alpha)$ of residues (distance = 8 Å, lower bound correction = 8 Å, upper-bound correction = 4 Å) to drive the protein–protein docking. All other parameters of HADDOCK were set as the default parameters. In each protein–protein docking, HADDOCK output 200 predicted complex structures ranked by the HADDOCK scores. As a control, we also performed protein–protein docking with HADDOCK in ab initio docking mode (center of mass restraints and random ambiguous interaction restraints definition). Besides, for homomeric PPIs, we additionally added the C2 symmetry constraint in both cases.

As shown in *Figure 7a and c*, the success rate of docking on homodimer and heterodimer test sets can be significantly improved when using the PLMGraph-Inter predicted inter-protein contacts as restraints. Whereas on the Homodimer test set, the success rate $(DockQ \geq 0.23)$ of the top 1 (top 10) prediction of HADDOCK in ab initio docking mode is 15.37% (39.18%), and when predicted by HADDOCK with PLMGraph-Inter predicted contacts, the success rate of the top 1 (top 10) prediction is 57.58% (61.04%). On the heterodimer test set, the success rate of top 1 (top 10) predictions of HADDOCK in ab initio docking mode is only 1.72% (6.32%), and when predicted by HADDOCK with PLMGraph-Inter predicted contacts, the success rate of top 1 (top 10) prediction is 29.89% (37.93%). From *Figure 7a and b* we can also see that integrating PLMGraph-Inter predicted contacts with HADDOCK not only allows for a higher success rate, but also more high-quality models in the docking results.

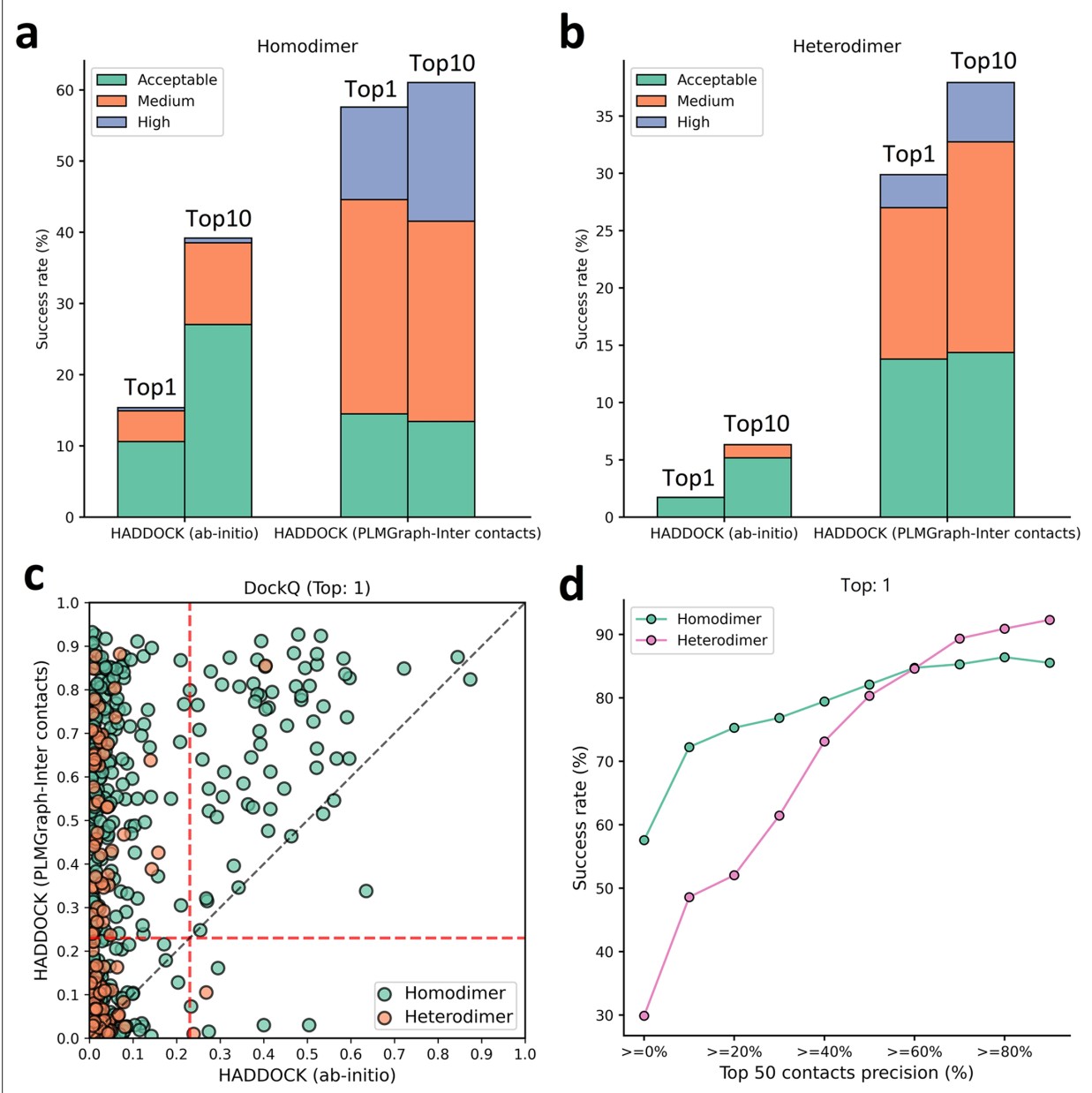

**Figure 7.** Protein–protein docking performances on the homodimer and heterodimer test sets. (**a, b**) The protein–protein docking performance comparison between HADDOCK with and without (ab initio) using PLMGraph-Inter predicted contacts as restraints on (**a**) homodimer and (**b**) heterodimer. The left side of each column shows the performance when the top 1 predicted model for each protein–protein interaction (PPI) is considered, and the right side shows the performance when the top 10 predicted models for each PPI are considered. (**c**) The head-to-head comparison of qualities of the top 1 model predicted by HADDOCK with and without using PLMGraph-Inter predicted contacts as restraints for each target PPI. The red lines represent the threshold (DockQ = 0.23) to determine whether the complex structure prediction is successful or not. (**d**) The success rates (the top 1 model) for protein complex structure prediction when only including targets for which precisions of the predicted contacts are higher than certain thresholds.

The online version of this article includes the following figure supplement(s) for figure 7:

**Figure supplement 1.** 3D structure of the homodimer (PDB: 3DFU).

**Figure supplement 2.** The comparison of HADDOCK (with PLMGraph-Inter contact constraints) with AlphaFold-Multimer in protein complex structure prediction.

We further explored the relationship between the precision of the top 50 contacts predicted by PLMGraph-Inter and the success rate of the top prediction of HADDOCK with PLMGraph-Inter predicted contacts. It can be clearly seen from *Figure 7d* that the success rate of protein–protein docking increases with the precision of contact prediction. Especially, when the precision of the predicted contacts reaches 50%, the docking success rate of both homologous and heterologous complexes can reach 80%, which is consistent with our finding in AlphaFold-Multimer. Therefore, we think this threshold can be used as a critical criterion for inter-protein contact prediction. It is important to emphasize that for some targets, although precisions of predicted contacts are very high, HADDOCK still failed to produce acceptable models. We manually checked these targets and found that many of these targets have at least one chain totally entangled by another chain (e.g., PDB 3DFU in *Figure 7—figure supplement 1*). We think large structural rearrangements may exist in forming the complex structures, which is difficult to model by traditional protein–protein docking approach.

Finally, we compared the qualities of the complex structures predicted by HADDOCK (with PLMGraph-Inter predicted contacts) and AlphaFold-Multimer. Although for some targets (e.g., PDB 5HPS in *Figure 7—figure supplement 2a*), the qualities of the structures predicted by HADDOCK were higher than those by AlphaFold-Multimer, for most targets, AlphaFold-Multimer generated higher quality structures (*Figure 7—figure supplement 2b and c*). Several reasons can account for the performance gap. First, precisions of the PLMGraph-Inter predicted contacts are still not enough, especially for heteromeric PPIs; second, HADDOCK cannot model large structural rearrangements in protein–protein docking, as we can see that for some targets, HADDOCK made poor predictions with high precisions of contact constraints (*Figure 7d*); and third, it is difficult to provide an objective evaluation of the true performance of AlphaFold-Multimer since many targets have already been included in the training set of AlphaFold-Multimer.

## Discussion

In this study, we proposed a new method to predict inter-protein contacts, denoted as PLMGraph-Inter. PLMGraph-Inter is based on the SE(3) invariant geometric graphs obtained from the structures of interacting proteins which are embedded with multiple PLMs. The predicted inter-protein contacts are obtained by successively transforming the PLM-embedded geometric graphs with graph encoders and residual networks. Benchmarking results on four test datasets show that PLMGraph-Inter outperforms five state-of-the-art inter-protein contact prediction methods including GLINTER, DeepHomo, CDPred, DeepHomo2, and DRN-1D2D_Inter by large margins, regardless of whether the experimental or predicted monomeric structures are used in building the geometric graphs. The ablation study further shows that the integration of the PLMs with the protein geometric graphs can dramatically improve the model performance, illustrating the efficacy of the PLM-embedded geometric graphs in protein representations. The protein representation framework proposed in this work can also be used to develop models for other tasks like protein function prediction, PPI prediction, etc. Recently, Fang et al. have also shown in their work that the incorporation of PLMs in geometric networks can significantly improve the model performances on a variety of protein-related tasks, including protein–protein interface prediction, model quality assessment, protein–protein rigid docking, and binding affinity prediction (*Wu et al., 2023*), which further supports this claim. We further show that PLMGraph-Inter can complement the result of AlphaFold-Multimer and leveraging the inter-protein contacts predicted by PLMGraph-Inter as constraints in protein–protein docking implemented with HADDOCK can dramatically improve its performance for protein complex structure prediction.

We noticed that although PLMGraph-Inter has achieved remarkable progress in inter-protein contact prediction, there is still room for further improvement, especially for heteromeric PPIs. Using more advanced PLMs, larger training datasets, and explicitly integrating physicochemical features of interacting proteins are directions worthy of exploration. Besides, since protein–protein docking approach generally has difficulties in modeling large conformational changes in PPIs, developing new approaches to integrate the predicted inter-protein contacts in the more advanced folding-and-docking framework like AlphaFold-Multimer or directly incorporating an additional structural module for protein complex structure generation in the network architecture can also be the future research directions.

# Methods

## Training and test datasets

We used the training set and test sets prepared in our previous work DRN-1D2D_Inter (*Si and Yan, 2023*) to train and evaluate PLMGraph-Inter. More details for the dataset generation can be found in the previous work. Specifically, we first prepared a non-redundant PPI dataset containing 4828 homomeric PPIs and 3134 heteromeric PPIs (with sequence identity 40% as the threshold), and after randomly selecting 400 homomeric PPIs (denoted as HomoPDB) and 200 heteromeric PPIs (denoted as HeteroPDB) as independent test sets, the remaining 7362 homomeric and heteromeric PPIs were used for training and validation.

DHTest and DB5.5 were also prepared in the work of DRN-1D2D_Inter by removing PPIs that are redundant (with sequence identity 40% as the threshold) to our training and validation set from the test set of DeepHomo and Docking Benchmark 5.5. DHTest contains 130 homomeric PPIs, and DB5.5 contains 59 heteromeric PPIs. Therefore, all the test sets used in this study are non-redundant (with sequence identity 40% as the threshold) to the dataset for the model development.

## Inter-protein contact definition

For a given PPI, two residues from the two interacting proteins are defined to be in contact if the distance of any two heavy atoms belonging to the two residues is smaller than 8 Å.

## Preparing the input features

### Geometric graphs from structures of interacting monomers

We first represent the protein as a graph, where each residue is represented as a node, and an edge is defined if the $C\alpha$ atom distance between two residues is less than 18 Å. (In our small-scale tests, increasing the cutoff used for defining edges can slightly increase the performance of the model. However, due to GPU memory limitations, we set the cutoff as 18 Å.) For each node and edge, we use scalars and vectors extracted from the 3D structures as their geometric features.

For each residue, we use its C,O,$C\alpha$,N, and a virtual $C\beta$ atom coordinates to extract information, the virtual $C\beta$ coordinates are calculated using the following formula (*Dauparas et al., 2022*): b = $C\alpha$ – N, c = C – $C\alpha$, a = cross(b, c), $C\beta$ = –0.58273431 * a + 0.56802827 * b – 0.54067466 * c + $C\alpha$.

To achieve an SE(3) invariant graph representation, as shown in *Figure 1—figure supplement 1b*, we define a local coordinate system on each residue (*Jumper et al., 2021*; *Pagès et al., 2019*). Specifically, for each residue, the unit vector in the $C\alpha - C$ direction is set as the $\vec{x}$ axis, the unit vector in the $C\alpha$-C-N plane and perpendicular $\vec{x}$ to is used as $\vec{y}$, and the z-direction is obtained through the cross-product of $\vec{x}$ and $\vec{y}$.

For the $i$th node, we use the three dihedral angles (ɸ, $\phi$, $\omega$) of the corresponding residue as the scalar features of the node (*Figure 1—figure supplement 1a*), and the unit vectors between the $C_i$, $N_i$, $O_i$, $C\alpha_i$, and C$\beta_i$ atoms of the corresponding residue and the $C_{i-1}$, $N_{i-1}$, $O_{i-1}$, $C\alpha_{i-1}$, and C$\beta_{i-1}$ atoms of the forward residue and the $C_{i+1}$, $N_{i+1}$, $O_{i+1}$, $C\alpha_{i+1}$, and C$\beta_{i+1}$ atoms of backward residue as the vector features of the node. In total, for each node, the dimension of the scalar features is 6 (each dihedral angle is encoded with its sine and cosine) and the dimension of the vector features is 50 *3.

For the edge between $i$th node and $j$th node, we use the distances and directions between the atoms of the two residues as the scalar features and vector features (see *Figure 1—figure supplement 1c*). The distances between the $C_i$, $N_i$, $O_i$, $C\alpha_i$, and C$\beta_i$ atoms of $i$th residue and the $C_j$, $N_j$, $O_j$, $C\alpha_j$, and C$\beta_j$ atoms of the $j$th residue are used as scalar features after encoded with the 16 Gaussian radial basis functions (*Jing et al., 2021b*). The position difference between $i$ and $j$ ($j$ – $i$) is also used as a scalar feature after sinusoidal encoding (*Vaswani et al., 2017*). The unit vectors between the $C_i$, $N_i$, $O_i$, $C\alpha_i$, and C$\beta_i$ atoms of $i$th residue and the $C_j$, $N_j$, $O_j$, $C\alpha_j$, and C$\beta_j$ atoms of the $j$th residue are used as vector features. In total, for each edge, the dimension of the scalar features is 432 and the dimension of the vector features is 25 * 3.

### Embeddings of single sequence, MSA, and structure

The single-sequence embedding is obtained by feeding the sequence into ESM-1b, and the structure embedding is obtained by feeding the structure into ESM-IF. To obtain the MSA embedding, we first search the Uniref100 protein sequence database for the sequence using JACKHMMER (*Potter*

*et al., 2018*) with the parameter (--incT L/2) to obtain the MSA, which is then inputted to hhmake (*Steinegger et al., 2019*) to get the HMM file, and to the LoadHMM.py script from RaptorX_Contact (*Wang et al., 2017*) to obtain the PSSM. The number of sequences of MSA is limited to 256 by hhfilter (*Steinegger et al., 2019*) and then input to ESM-MSA-1b to get the MSA embedding. The dimensions of the sequence embedding, PSSM, MSA embedding, and structural embeddings are 1280, 20, 768, and 512, respectively. After adding embeddings to the scalar features of the nodes, the dimension of the scalar features of each node is 2586.

## 2D feature from paired MSA

For homomeric PPIs, the paired MSA is formed by concatenating two copies of the MSA. For heteromeric PPIs, the paired MSA is formed by pairing the MSAs through the phylogeny-based approach described in https://github.com/ChengfeiYan/PPI_MSA-taxonomy_rank (*Yan, 2024*; *Si and Yan, 2022*). We input the paired MSA into CCMpred (*Seemayer et al., 2014*) to get the evolutionary coupling matrix and into alnstats (*Jones et al., 2015*) to get mutual information matrix, APC-corrected mutual information matrix, and contact potential matrix. The number of sequences of paired MSA is limited to 256 by hhfilter (*Steinegger et al., 2019*) and then input to ESM-MSA-1b to get the attention maps. In total, the channel of 2D features is 148.

## GVP and GVPConv

GVP is a two-track neural network module consisting of a scalar track and a vector track, which can perform SE(3) invariant transformations on scalar features and SE(3) equivariant transformations on vector features. A detailed description can be found in the work of GVP (*Jing et al., 2021a*; *Jing et al., 2021b*).

GVPConv is a message passing-based graph neural network, which mainly consists of a message function and a feedforward function. Where the message function contains a sequence of three GVP modules and the feedforward function contains a sequence of two GVP modules. GVPConv is used to transform the node features. Specifically, the input node features are first processed by the message function. We denote the features of node $i$ by $h^i$, the feature of edge $(j \rightarrow i)$ by $h^{j \rightarrow i}$, the set of nodes connected to node $i$ by $\varepsilon_i$, and the three GVP modules of the message function by $gm$, then the node features processed by the message function can be represented as

$$h^i_m = \frac{1}{len\left(\varepsilon_i\right)} \left( \sum_{j \in \varepsilon_i} gm \left( concat \left( h^i, h^{j \rightarrow i}, h^j \right) \right) \right) \tag{1}$$

where $len\left(\varepsilon_i\right)$ denotes the number of nodes connected to node $i$. After sequential normalization (*Equation 2*) and feedforward function (*Equation 3*), the features of node $i$ updated by GVPConv Layer are obtained:

$$h^i_m = \text{LayerNorm}\left(h^i_m + \text{Dropout}\left(h^i_m\right)\right) \tag{2}$$

$$h^i_{out} = \text{LayerNorm}\left(h^i_m + \text{Dropout}\left(gs\left(h^i_m\right)\right)\right) \tag{3}$$

where $gs$ denotes the two GVP modules of the feedforward function, and $h^i_{out}$ denotes the outputs of GVPConv Layer.

## The transforming procedure in the residual network module

We first use a convolution layer with kernel size of 1 * 1 to reduce the number of channels of the input 2D feature maps from 1044 to 96, which are then transformed successively by nine-dimensional hybrid residual blocks and another convolution layer with kernel size of 1 * 1 for the channel reduction (from 96 to 1). Finally, we use the sigmoid function to transform the feature map to obtain the predicted inter-protein contact map.

## Training protocol

Our training set contains 7362 PPIs, and we used sevenfold cross-validation to train PLMGraph-Inter. Specifically, we randomly divided the training set into seven subsets, and each time, we selected

six subsets as the training set and the remaining subset as the validation set. Seven models were trained in total, and the final prediction was the average of the predictions from the seven models. Each model was trained using AdamW optimizer with 0.001 as the initial learning rate, in which the singularity enhanced loss function proposed by us in our previous study (*Si and Yan, 2021*) was used to calculate the training and validation loss. During training, if the validation loss did not decrease within two epochs, we would decay the learning rate by 0.1. The training stopped after the learning rate decayed twice and the model with the highest top 50 mean precision on the validation dataset was saved as the prediction model.

PLMGraph-Inter was implemented with pytorch (v.1.11) and trained on one NVIDIA TESLA A100 GPU with batch size equaling 1. Due to memory limitation of GPU, the length of each protein sequence was limited to 400. When a sequence was longer than 400, a fragment with sequence length equaling 400 was randomly selected in the model training.

## Quality assessment of the predicted protein complex structures

We evaluated the models generated by AlphaFold-Multimer and HADDOCK using DockQ (*Basu and Wallner, 2016*), a score ranging between 0 and 1. Specifically, a model with DockQ < 0.23 means that the prediction is incorrect; $0.23 \leq$ DockQ < 0.49 means the model is an acceptable prediction; $0.49 \leq$ DockQ < 0.8 corresponds to a medium-quality prediction; and $0.8 \leq$ DockQ corresponds to a high-quality prediction.

## Further potential redundancies removal between the training and the test sets

### Removing potential redundancies using different sequence similarity thresholds

CD-HIT (*Li et al., 2001*) was originally used in removing redundancies between the training and test sets used in this study. Since the lowest sequence identity threshold accepted by CD-HIT is 40%, to use more stringent threshold in the redundancy removal, we further clustered all the monomer sequences from the training set and the test sets (HomoPDB, HeteroPDB) using MMSeq2 (*Steinegger and Söding, 2018*) with different sequence identity thresholds (40%, 30%, 20%, 10%). Under a certain threshold, each sequence is uniquely labeled by the cluster (e.g., cluster 0, cluster 1, …) to which it belongs, from which each PPI can be marked with a pair of clusters (e.g., cluster 0–cluster 1). The PPIs belonging to the same cluster pair (note: cluster *n* – cluster *m* and cluster *n* – cluster *m* were considered as the same pair) were considered redundant with this sequence identity threshold. For each PPI in the test set, if the pair cluster it belongs to contains any PPI belonging to the training set, we remove that PPI from the test set.

### Removing potential redundancies using different fold similarity thresholds of interacting monomers

We used TM-align (*Zhang and Skolnick, 2005*) to evaluate the fold similarities (in TM-scores) between the experimental structures of the interacting monomers in the training set and the test sets (HomoPDB, HeteroPDB). Specifically, for any two targets *A–B* and *A'–B'* in the training set and test sets, respectively, where *A, B, A'*, and *B'* represent the interacting monomers. We calculated the MTM-score defined as

$$\text{MTM-score} = \max \begin{cases} \min \left( TM - score_{AA'}, TM - score_{BB'} \right) \\ \min \left( TM - score_{AB'}, TM - score_{BA'} \right) \end{cases} \tag{4}$$

between the two targets. The MTM-score higher than a certain value means that both the two interacting monomers in the two targets have fold similarity scores (TM-scores) higher than this value. When a threshold is chosen, we remove targets in the test tests if they have MTM-scores higher than this threshold when compared with any target in the training set. In this study, different thresholds, including 0.9, 0.8, 0.7, 0.6, and 0.5, were used in the study. 0.5 was chosen as the lowest threshold for protein pairs with TM-score < 0.5 mainly not in the same fold.

## Calculating the normalized number of the effective sequences of paired MSA

We define the normalized number of the effective sequences ($N_{eff}^{norm}$) as follows:

$$N_{eff}^{norm} = \frac{1}{\sqrt{L}} \sum_{n=1}^{N} \frac{1}{\sum_{m=1}^{N} I\left[S_{m,n} \geq 0.8\right]} \tag{5}$$

where $L$ is the length of the paired MSA, $N$ is the number of sequences in the paired MSA, $S_{m,n}$ is the sequence identity between the $m$th and $n$th sequences, and $I[\ ]$ represents the Iverson bracket, which means $I\left[S_{m,n} \geq 0.8\right] = 1$ if $S_{m,n} \geq 0.8$ or 0 otherwise.

## Acknowledgements

The work was supported by the National Natural Science Foundation of China (32101001) and a new faculty startup grant (3004012167) from the Huazhong University of Science and Technology. The computation was completed in the HPC Platform of Huazhong University of Science and Technology.

## Additional information

### Funding

| Funder | Grant reference number | Author |
|---|---|---|
| National Natural Science Foundation of China | 32101001 | Chengfei Yan |
| Huazhong University of Science and Technology | 3004012167 | Chengfei Yan |

The funders had no role in study design, data collection and interpretation, or the decision to submit the work for publication.

### Author contributions

Yunda Si, Conceptualization, Resources, Data curation, Software, Formal analysis, Validation, Investigation, Visualization, Methodology, Writing - original draft, Writing - review and editing; Chengfei Yan, Conceptualization, Resources, Data curation, Software, Formal analysis, Supervision, Funding acquisition, Validation, Investigation, Visualization, Methodology, Writing - original draft, Project administration, Writing - review and editing

### Author ORCIDs

Chengfei Yan http://orcid.org/0000-0002-2010-6668

Reviewer #1 (Public Review): https://doi.org/10.7554/eLife.92184.3.sa1
Reviewer #2 (Public Review): https://doi.org/10.7554/eLife.92184.3.sa2
Author Response https://doi.org/10.7554/eLife.92184.3.sa3

## Additional files

### Supplementary files

• Supplementary file 1. The performances of DeepHomo, GLINTER, DRN-1D2D_Inter, DeepHomo2, CDPred, and PLMGraph-Inter on HomoPDB and HeteroPDB after the removal of targets which GLINTER failed to make the prediction using experimental structures (AlphaFold2 predicted structures).

• Supplementary file 2. The performances of different ablation study models on the HomoPDB and HeteroPDB test sets.

• Supplementary file 3. The performances of DeepHomo, GLINTER, DRN-1D2D_Inter, DeepHomo2,

CDPred, and PLMGraph-Inter on the DHTest and DB5.5 test sets using experimental structures (AlphaFold2 predicted structures).

• Supplementary file 4. The performances of DeepHomo, GLINTER, DRN-1D2D_Inter, DeepHomo2, CDPred, and PLMGraph-Inter on DHTest and DB5.5 after the removal of targets which GLINTER failed to make the prediction using experimental structures (AlphaFold2 predicted structures).

• Supplementary file 5. The performances of AlphaFold-Multimer and PLMGraph-Inter on the homodimer and heterodimer test sets.

• MDAR checklist

## Data availability

The current manuscript contains results which are computational. These are all presented and discussed in the main text. The datasets for training and testing PLMGraph-Inter are provided in https://github.com/ChengfeiYan/PLMGraph-Inter/tree/main/data. The code for training and implementing PLMGraph-Inter is provided in https://github.com/ChengfeiYan/PLMGraph-Inter (copy archived at *Yan, 2023*). The code for implementing PLMGraph-Inter is provided in https://github.com/ChengfeiYan/PLMGraph-Inter.

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
