## [Editor Report · eLife assessment]

This study presents a **useful** deep learning-based inter-protein contact prediction method named PLMGraph-Inter which combines protein language models and geometric graphs. The evidence supporting the claims of the authors is **solid**. The authors show that their approach may be used in cases where AlphaFold-Multimer performs poorly. This work will be of interest to researchers working on protein complex structure prediction, particularly when accurate experimental structures are available for one or both of the monomers in isolation.

---

## [Referee Report · Reviewer #1 (Public Review)]

Summary:

Given knowledge of the amino acid sequence and of some version of the 3D structure of two monomers that are expected to form a complex, the authors investigate whether it is possible to accurately predict which residues will be in contact in the 3D structure of the expected complex. To this effect, they train a deep learning model which takes as inputs the geometric structures of the individual monomers, per-residue features (PSSMs) extracted from MSAs for each monomer, and rich representations of the amino acid sequences computed with the pre-trained protein language models ESM-1b, MSA Transformer, and ESM-IF. Predicting inter-protein contacts in complexes is an important problem. Multimer variants of AlphaFold, such as AlphaFold-Multimer, are the current state of the art for full protein complex structure prediction, and if the three-dimensional structure of a complex can be accurately predicted then the inter-protein contacts can also be accurately determined. By contrast, the method presented here seeks state-of-the-art performance among models that have been trained end-to-end for inter-protein contact prediction.

Strengths:

The paper is carefully written and the method is very well detailed. The model works both for homodimers and heterodimers. The ablation studies convincingly demonstrate that the chosen model architecture is appropriate for the task. Various comparisons suggest that PLMGraph-Inter performs substantially better, given the same input, than DeepHomo, GLINTER, CDPred, DeepHomo2, and DRN-1D2D_Inter.

The authors control for some degree of redundancy between their training and test sets, both using sequence and structural similarity criteria. This is more careful than can be said of most works in the field of PPI prediction.

As a byproduct of the analysis, a potentially useful heuristic criterion for acceptable contact prediction quality is found by the authors: namely, to have at least 50% precision in the prediction of the top 50 contacts.

Weaknesses:

The authors check for performance drops when the test set is restricted to pairs of interacting proteins such that the chain pair is not similar *as a pair* (in sequence or structure) to a pair present in the training set. A more challenging test would be to restrict the test set to pairs of interacting proteins such that *none* of the chains are separately similar to monomers present in the training set. In the case of structural similarity (TM-scores), this would amount to replacing the two "min"s with "max"s in Eq. (4). In the case of sequence similarity, one would simply require that no monomer in the test set is in any MMSeqs2 cluster observed in the training set. This may be an important check to make, because a protein may interact with several partners, and/or may use the same sites for several distinct interactions, contributing to residual data leakage in the test set.

The training set of AFM with v2 weights has a global cutoff of 30 April 2018, while that of PLMGraph-Inter has a cutoff of March 7 2022. So there may be structures in the test set for PLMGraph-Inter that are not in the training set of AFM with v2 weights (released between May 2018 and March 2022). The "Benchmark 2" dataset from the AFM paper may have a few additional structures not in the training or test set for PLMGraph-Inter. I realize there may be only few structures that are in neither training set, but still think that showing the comparison between PLMGraph-Inter and AFM there would be important, even if no statistically significant conclusions can be drawn.

Finally, the inclusion of AFM confidence scores is very good. A user would likely trust AFM predictions when the confidence score is high, but look for alternative predictions when it is low. The authors' analysis (Figure 6, panels c and d) seems to suggest that, in the case of heterodimers, when AFM has low confidence, PLMGraph-Inter improves precision by (only) about 3% on average. By comparison, the reported gains in the "DockQ-failed" and "precision-failed" bins are based on knowledge of the ground truth final structure, and thus are not actionable in a real use-case.

---

## [Referee Report · Reviewer #2 (Public Review)]

This work introduces PLMGraph-Inter, a new deep learning approach for predicting inter-protein contacts, which is crucial for understanding protein-protein interactions. Despite advancements in this field, especially driven by AlphaFold, prediction accuracy and efficiency in terms of computational cost still remains an area for improvement. PLMGraph-Inter utilizes invariant geometric graphs to integrate the features from multiple protein language models into the structural information of each subunit. When compared against other inter-protein contact prediction methods, PLMGraph-Inter shows better performance which indicates that utilizing both sequence embeddings and structural embeddings is important to achieve high-accuracy predictions with relatively smaller computational costs for the model training.

---

## [Author Response]

The following is the authors’ response to the current reviews.

Overall Response

We thank the reviewers for reviewing our manuscript, recognizing the significance of our study, and offering valuable suggestions. Based on the reviewer’s comments and the updated eLife assessment, we would like to chose the current version of our manuscript as the Version of Record of our manuscript.

**Public Reviews:**

**Reviewer #1 (Public Review):**
Summary:Given knowledge of the amino acid sequence and of some version of the 3D structure of two monomers that are expected to form a complex, the authors investigate whether it is possible to accurately predict which residues will be in contact in the 3D structure of the expected complex. To this effect, they train a deep learning model which takes as inputs the geometric structures of the individual monomers, per-residue features (PSSMs) extracted from MSAs for each monomer, and rich representations of the amino acid sequences computed with the pre-trained protein language models ESM-1b, MSA Transformer, and ESM-IF. Predicting inter-protein contacts in complexes is an important problem. Multimer variants of AlphaFold, such as AlphaFold-Multimer, are the current state of the art for full protein complex structure prediction, and if the three-dimensional structure of a complex can be accurately predicted then the inter-protein contacts can also be accurately determined. By contrast, the method presented here seeks state-of-the-art performance among models that have been trained end-to-end for inter-protein contact prediction.Strengths:The paper is carefully written and the method is very well detailed. The model works both for homodimers and heterodimers. The ablation studies convincingly demonstrate that the chosen model architecture is appropriate for the task. Various comparisons suggest that PLMGraph-Inter performs substantially better, given the same input, than DeepHomo, GLINTER, CDPred, DeepHomo2, and DRN-1D2D_Inter.The authors control for some degree of redundancy between their training and test sets, both using sequence and structural similarity criteria. This is more careful than can be said of most works in the field of PPI prediction.As a byproduct of the analysis, a potentially useful heuristic criterion for acceptable contact prediction quality is found by the authors: namely, to have at least 50% precision in the prediction of the top 50 contacts.

We thank the reviewer for recognizing the strengths of our work!

Weaknesses:The authors check for performance drops when the test set is restricted to pairs of interacting proteins such that the chain pair is not similar *as a pair* (in sequence or structure) to a pair present in the training set. A more challenging test would be to restrict the test set to pairs of interacting proteins such that *none* of the chains are separately similar to monomers present in the training set. In the case of structural similarity (TM-scores), this would amount to replacing the two "min"s with "max"s in Eq. (4). In the case of sequence similarity, one would simply require that no monomer in the test set is in any MMSeqs2 cluster observed in the training set. This may be an important check to make, because a protein may interact with several partners, and/or may use the same sites for several distinct interactions, contributing to residual data leakage in the test set.

We thank the reviewer for the suggestion! In the case of protein-protein prediction (“0D prediction”) or protein-protein interfacial residue prediction(“1D prediction”), we think making none of the chains in the test set separately similar to monomers in the training set is necessary, as the reviewer pointed out that a protein may interact with several partners, and may even use the same sites for the interactions. Since the task of this study is predicting the inter-protein residue-residue contacts (“2D prediction”), even though a protein uses the same site to interact with different partners, as long as the interacting partners are different, the inter-protein contact maps would be different. Therefore, we don’t think that in our task, making this restriction to the test set is necessary.

The training set of AFM with v2 weights has a global cutoff of 30 April 2018, while that of PLMGraph-Inter has a cutoff of March 7 2022. So there may be structures in the test set for PLMGraph-Inter that are not in the training set of AFM with v2 weights (released between May 2018 and March 2022). The "Benchmark 2" dataset from the AFM paper may have a few additional structures not in the training or test set for PLMGraph-Inter. I realize there may be only few structures that are in neither training set, but still think that showing the comparison between PLMGraph-Inter and AFM there would be important, even if no statistically significant conclusions can be drawn.

We thank the reviewer for the suggestion! It is not enough to only use the date cutoff to remove the redundancy, since similar structures can be deposited in the PDB in different dates. Because AFM does not release the PDB codes of its training set, it is difficult for us to totally remove the redundancy. Therefore, we think no rigorous conclusion can be drawn by including these comparisons in the manuscript. Besides, the main point of this study is to demonstrate that the integration of multiple protein language models using protein geometric graphs can dramatically improve the model performance for inter-protein contact prediction, which can provide some important enlightenments for the future development of more powerful protein complex structure prediction methods beyond AFM, rather than providing a tool which can beat AFM at this moment. We think including too many stuffs in the comparison with AFM may distract the readers. Therefore, we choose to not include these comparisons in the manuscript.

Finally, the inclusion of AFM confidence scores is very good. A user would likely trust AFM predictions when the confidence score is high, but look for alternative predictions when it is low. The authors' analysis (Figure 6, panels c and d) seems to suggest that, in the case of heterodimers, when AFM has low confidence, PLMGraph-Inter improves precision by (only) about 3% on average. By comparison, the reported gains in the "DockQ-failed" and "precision-failed" bins are based on knowledge of the ground truth final structure, and thus are not actionable in a real use-case.

We agree with the reviewer that more studies are needed for providing a model which can well complement or even beat AFM. The main point of this study is to demonstrate that the integration of multiple protein language models using protein geometric graphs can dramatically improve the model performance for inter-protein contact prediction, which can provide some important enlightenments for the future development of more powerful protein complex structure prediction methods beyond AFM.

**Reviewer #2 (Public Review):**
This work introduces PLMGraph-Inter, a new deep learning approach for predicting inter-protein contacts, which is crucial for understanding proteinprotein interactions. Despite advancements in this field, especially driven by AlphaFold, prediction accuracy and efficiency in terms of computational cost still remains an area for improvement. PLMGraph-Inter utilizes invariant geometric graphs to integrate the features from multiple protein language models into the structural information of each subunit. When compared against other inter-protein contact prediction methods, PLMGraph-Inter shows better performance which indicates that utilizing both sequence embeddings and structural embeddings is important to achieve high-accuracy predictions with relatively smaller computational costs for the model training.

We thank the reviewer for recognizing the strengths of our work!

**Recommendations for the authors:**

**Reviewer #1 (Recommendations For The Authors):**
I recommend renaming the section "Further potential redundancies removal between the training and the test" to "Further potential redundancies removal between the training and the test sets"

Changed.

In lines 768-769, the sentence seems to end prematurely in "to use more stringent threshold in the redundancy removal"

Corrected.

In Eq. (4), line 789, there are many instances of dashes that look like minus signs, creating some confusion.

Corrected.

I think I may have mixed up figure references in my first review. When I said (Recommendations to the authors): "p. 22, line 2: from the figure, I would have guessed "greater than or equal to 0.7", not 0.8", I think I was referring to what is now lines 423-424, referring to what is now Figure 5c. The point stands there, I think.

Corrected.

A couple of new grammatical mishaps have been introduced in the revision. These could be rectified.

We carefully rechecked our revisions, and corrected the grammatical issues we found.

**Reviewer #2 (Recommendations For The Authors):**
Most of my concerns were resolved through the revision. I have only one suggestion for the main figure.The current scatter plots in Figure 2 are hard to understand as too many different methods are abstracted into a single plot with multiple colors. I would suggest comparing their performances using box plot or violin plot for the figure 2.

We thank the reviewer for the suggestion! In the revision, we tried violin plot, but it does not look good since too many different methods are included in the plot. Besides, we chose the scatter plot as it can provide much more details. We also provided the individual head-to-head scatter plots as supplementary figures, we think which can also be helpful for the readers to capture the information of the figures.

The following is the authors’ response to the original reviews.

Overall Response

We would like to thank the reviewers for reviewing our manuscript, recognizing the significance of our study, and offering valuable suggestions. We have carefully revised the manuscript to address all the concerns and suggestions raised by the reviewers.

**Public Reviews:**

**Reviewer #1 (Public Review):**
Summary:Given knowledge of the amino acid sequence and of some version of the 3D structure of two monomers that are expected to form a complex, the authors investigate whether it is possible to accurately predict which residues will be in contact in the 3D structure of the expected complex. To this effect, they train a deep learning model that takes as inputs the geometric structures of the individual monomers, per-residue features (PSSMs) extracted from MSAs for each monomer, and rich representations of the amino acid sequences computed with the pre-trained protein language models ESM-1b, MSA Transformer, and ESM-IF. Predicting inter-protein contacts in complexes is an important problem. Multimer variants of AlphaFold, such as AlphaFold-Multimer, are the current state of the art for full protein complex structure prediction, and if the three-dimensional structure of a complex can be accurately predicted then the inter-protein contacts can also be accurately determined. By contrast, the method presented here seeks state-of-the-art performance among models that have been trained end-to-end for inter-protein contact prediction.Strengths:The paper is carefully written and the method is very well detailed. The model works both for homodimers and heterodimers. The ablation studies convincingly demonstrate that the chosen model architecture is appropriate for the task. Various comparisons suggest that PLMGraph-Inter performs substantially better, given the same input than DeepHomo, GLINTER, CDPred, DeepHomo2, and DRN-1D2D_Inter. As a byproduct of the analysis, a potentially useful heuristic criterion for acceptable contact prediction quality is found by the authors: namely, to have at least 50% precision in the prediction of the top 50 contacts.

We thank the reviewer for recognizing the strengths of our work!

Weaknesses:My biggest issue with this work is the evaluations made using *bound* monomer structures as inputs, coming from the very complexes to be predicted.Conformational changes in protein-protein association are the key element of the binding mechanism and are challenging to predict. While the GLINTER paper (Xie & Xu, 2022) is guilty of the same sin, the authors of CDPred (Guo et al., 2022) correctly only report test results obtained using predicted unbound tertiary structures as inputs to their model. Test results using experimental monomer structures in bound states can hide important limitations in the model, and thus say very little about the realistic use cases in which only the unbound structures (experimental or predicted) are available. I therefore strongly suggest reducing the importance given to the results obtained using bound structures and emphasizing instead those obtained using predicted monomer structures as inputs.

We thank the reviewer for the suggestion! In the revision, to emphasize the performance of PLMGraph-Inter using the predicted monomer structures, we moved the evaluation results based on the predicted monomer from the supplementary to the main text (see the new Table 1 and Figure 2 in the revised manuscript) and re-organized the two subsections “Evaluation of PLMGraph-Inter on HomoPDB and HeteroPDB test sets” and “Impact of the monomeric structure quality on contact prediction” in the main text.

In particular, the most relevant comparison with AlphaFold-Multimer (AFM) is given in Figure S2, *not* Figure 6. Unfortunately, it substantially shrinks the proportion of structures for which AFM fails while PLMGraph-Inter performs decently. Still, it would be interesting to investigate why this occurs. One possibility would be that the predicted monomer structures are of bad quality there, and PLMGraph-Inter may be able to rely on a signal from its language model features instead. Finally, AFM multimer confidence values ("iptm + ptm") should be provided, especially in the cases in which AFM struggles.

We thank the reviewer for the suggestion! It is worth noting that AFM automatically searches monomer templates in the prediction, and when we checked our AFM runs, we found that 99% of the targets in our study (including all the targets in the four datasets: HomoPDB, HeteroPDB, DHTest and DB5.5) at least 20 templates were identified (AFM employed the top 20 templates in the prediction), and 87.8% of the targets employed the native templates (line 455-462 in page 25 in the subsection of “Comparison of PLMGraph-Inter with AlphaFold-Multimer”). Therefore, we think Figure 6 not Figure S5 (the original Figure S2) shows a fairer comparison. Besides, it is also worth noting the targets used in this study would have a large overlap with the training set of AlphaFold-Multimer, since AFM used all protein complex structures in PDB deposited before 2018-04-30 in the model training, which would further cause the overestimation of the performance of AFM (line 450-455 in page 24-25 in the subsection of “Comparison of PLMGraph-Inter with AlphaFold-Multimer”).

To mimic the performance of AlphaFold2 in real practice and produce predicted monomeric structures with more diverse qualities, we only used the MSA searched from Uniref100 protein sequence database as the input to AlphaFold2 and set to not use the template (line 203~210 in page 12 in the subsection of “Evaluation of PLMGraph-Inter on HomoPDB and HeteroPDB test sets”). Since some of the predicted monomer structures are of bad quality, it is reasonable that the performance of PLMGraph-Inter drops when the predicted monomeric structures are used in the prediction. We provided a detailed analysis of the impact of the monomeric structure quality on the prediction performance in the subsection “Impact of the monomeric structure quality on contact prediction” in the main text.

We provided the analysis of the AFM multimer confidence values (“iptm + ptm”) in the revision (Figure 6, Figure S5 and line 495-501 in page 27 in the subsection of“Comparison of PLMGraph-Inter with AlphaFold-Multimer”).

Besides, in cases where *any* experimental structures - bound or unbound - are available and given to PLMGraph-Inter as inputs, they should also be provided to AlphaFold-Multimer (AFM) as templates. Withholding these from AFM only makes the comparison artificially unfair. Hence, a new test should be run using AFM templates, and a new version of Figure 6 should be produced. Additionally, AFM's mean precision, at least for top-50 contact prediction, should be reported so it can be compared with PLMGraph-Inter's.

We thank the reviewers for the suggestion, and we are sorry for the confusion! In the AFM runs to predict protein complex structures, we used the default setting of AFM which automatically searches monomer templates in the prediction. When we checked our AFM runs, we found that 99% of the targets in our study (including all the targets in the four datasets: HomoPDB, HeteroPDB, DHTest and DB5.5) employed at least 20 templates in their predictions (AFM only used the top 20 templates), and 87.8% of the targets employed the native template. We further clarified this in the revision (line 455462 in page 25 in the subsection of “Comparison of PLMGraph-Inter with AlphaFoldMultimer”). We also included the mean precisions of AFM (top-50 contact prediction) in the revision (Table S5 and line 483-484 in page 26 in the subsection of “Comparison of PLMGraph-Inter with AlphaFold-Multimer”).

It's a shame that many of the structures used in the comparison with AFM are actually in the AFM v2 training set. If there are any outside the AFM v2 training set and, ideally, not sequence- or structure-homologous to anything in the AFM v2 training set, they should be discussed and reported on separately. In addition, why not test on structures from the "Benchmark 2" or "Recent-PDB-Multimers" datasets used in the AFM paper?

We thank the reviewer for the suggestion! The biggest challenge to objectively evaluate AFM is that as far as we known, AFM does not release the PDB ids of its training set and the “Recent-PDB-Multimers” dataset. “Benchmark 2” only includes 17 heterodimer proteins, and the number would be further decreased after removing targets redundant to our training set. We think it is difficult to draw conclusions from such a small number of targets.

It is also worth noting that the AFM v2 weights have now been outdated for a while, and better v3 weights now exist, with a training cutoff of 2021-09-30.

**Author response image 1. sa3fig1:** The head-to-head comparison of qualities of complex predicted by AlphaFold-Multimer (2.2.0) and AlphaFold-Multimer (2.3.2) for each target PPI.

We thank the reviewer for reminding the new version of AFM. The only difference between AFM V3 and V2 is the cutoff date of the training set. During the revision, we also tested the new version of AFM on the datasets of HomoPDB and HeteroPDB, but we found the performance difference between the two versions of AFM is actually very little (see the figure above, not shown in the main text). One reason might be that some targets in HomoPDB and HeteroPDB are redundant with the training sets of the two version of AFM. Since our test sets would have more overlaps with the training set of AFM V3, we keep using the AFM V2 weights in this study.

Another weakness in the evaluation framework: because PLMGraph-Inter uses structural inputs, it is not sufficient to make its test set non-redundant in sequence to its training set. It must also be non-redundant in structure. The Benchmark 2 dataset mentioned above is an example of a test set constructed by removing structures with homologous templates in the AF2 training set. Something similar should be done here.

We thank the reviewer for the suggestion! In the revision, we explored the performance of PLMGraph-Inter when using different thresholds of fold similarity scores of interacting monomers to further remove potential redundancies between the training and test sets (i.e. redundancy in structure ) (line 353-386 in page 19-21 in the subsection “Ablation study”; line 762-797 in page 41-43 in the subsection “Further potential redundancies removal between the training and the test”). We found that for heteromeric PPIs (targets in HeteroPDB), the further removal of potential redundancy in structure has little impact on the model performance (~3%, when TM-score 0.5 is used as the threshold). However, for homomeric PPIs (targets in HomoPDB), the further removal of potential redundancy in structure significantly reduce the model performance (~18%, when TM-score 0.5 is used as the threshold) (see Table 2). One possible reason for this phenomenon is that the binding mode of the homomeric PPI is largely determined by the fold of its monomer, thus the does not generalize well on targets whose folds have never been seen during the training.

Whether the deep learning model can generalize well on targets with novel folds is a very interesting and important question. We thank the reviewer for pointing out this!However, to the best of our knowledge, this question has rarely been addressed by previous studies including AFM. For example, the Benchmark 2 dataset is prepared by ClusPro TBM (bioRxiv 2021.09.07.459290; Proteins 2020, 88:1082-1090) which uses a sequence-based approach (HHsearch) to identify templates not structure-based.Therefore, we don’t think this dataset is non-redundant in structure.

Finally, the performance of DRN-1D2D for top-50 precision reported in Table 1 suggests to me that, in an ablation study, language model features alone would yield better performance than geometric features alone. So, I am puzzled why model "a" in the ablation is a "geometry-only" model and not a "LM-only" one.

Using the protein geometric graph to integrate multiple protein language models is the main idea of PLMGraph-Inter. Comparing with our previous work (DRN-1D2D_Inter), we consider the building of the geometric graph as one major contribution of this work. To emphasize the efficacy of this geometric graph, we chose to use the “geometry-only” model as the base model.

**Reviewer #1 (Recommendations For The Authors):**
Some sections of the paper use technical terminology which limits accessibility to a broad audience. An obvious example is in the section "Results > Overview of PLMGraph-Inter > The residual network module": the average eLife reader is not a machine learning expert and might not be familiar with a "convolution with kernel size of 1 * 1". In general, the "Overview of PLMGraph-Inter" is a bit heavy with technical details, and I suggest moving many of these to Methods. This overview section can still be there but it should be shorter and written using less technical language.

We thank the reviewer for the suggestion! We moved some technical details to the Methods section in the revision (line 184-185 in page 11; line 729-735 in page 39).

List of typos and minor issues (page number according to merged PDF):p. 3. line -3: remove "to"

Corrected (line 36, page 3)

p. 5, line 7: "GINTER" should be "GLINTER"

Corrected (line 64, page 5)

p. 6, line -4: "Given structures" -> "Given the structures"

Corrected (line 95, page 6)

p. 6, line -2: "with which encoded"... ?

We rephrased this sentence in revision. (line 97, page 6)

p. 9, line 1: "principal" -> "principle"

Corrected (line 142, page 9)

p. 13, line 1: "has" -> "but have"

Corrected (line 231, page 13)

p. 14, lines 6-7: "As can be seen from the figure that the predicted" -> "As can be seen from the figure, the predicted"

We rephrased this paragraph, and the sentence was deleted in the revision (line 257-259 in page 15).

p. 18, line 1: the "five models" are presumably models a-e? If so, say "of models a-e"

Corrected (line 310, page 17)

p. 22, line 2: from the figure, I would have guessed "greater than or equal to 0.7", not 0.8

Based the Figure 3C, we think 0.8 is a more appropriate cutoff, since the precision drops significantly when the DTM-score is within 0.7~0.8.

p. 23, lines 2-3: "worth to making" -> "worth making"

Corrected (line 443, page 24)

p. 24, line -5: "predict" -> "predicted"

Corrected (line 484, page 26)

p 28, line -5: Please clarify what you mean by "We doubt": are you saying that you don't think these rearrangements exist in nature? If not, then reword.

Corrected (line 566, page 30)

Figure 2, panel c, "DCPred" in the legend should be "CDPred"

Corrected

Figures 3 and 5: Please improve the y-axis title in panel C. "Percent" of what?

We changed the “Percent” to “% of targets” in the revision.

We thank the reviewer for carefully reading our manuscript!

**Reviewer #2 (Public Review):**
This work introduces PLMGraph-Inter, a new deep-learning approach for predicting inter-protein contacts, which is crucial for understanding proteinprotein interactions. Despite advancements in this field, especially driven by AlphaFold, prediction accuracy and efficiency in terms of computational cost still remains an area for improvement. PLMGraph-Inter utilizes invariant geometric graphs to integrate the features from multiple protein language models into the structural information of each subunit. When compared against other inter-protein contact prediction methods, PLMGraph-Inter shows better performance which indicates that utilizing both sequence embeddings and structural embeddings is important to achieve high-accuracy predictions with relatively smaller computational costs for the model training.The conclusions of this paper are mostly well supported by data, but test examples should be revisited with a more strict sequence identity cutoff to avoid any potential information leakage from the training data. The main figures should be improved to make them easier to understand.

We thank the reviewer for recognizing the significance of our work! We have carefully revised the manuscript to address the reviewer’s concerns.

(1) The sequence identity cutoff to remove redundancies between training and test set was set to 40%, which is a bit high to remove test examples having homology to training examples. For example, CDPred uses a sequence identity cutoff of 30% to strictly remove redundancies between training and test set examples. To make their results more solid, the authors should have curated test examples with lower sequence identity cutoffs, or have provided the performance changes against sequence identities to the closest training examples.

We thank the reviewer for the valuable suggestion! The “40 sequence identity” is a widely used threshold to remove redundancy when evaluating deep-learning based protein-protein interaction and protein complex structure prediction methods, thus we also chose this threshold in our study (bioRxiv 2021.10.04.463034, Cell Syst. 2021 Oct 20;12(10):969-982.e6). In the revision, we explored whether PLMGraph-inter can keep its performance when more stringent thresholds (30%,20%,10%) is applied (line 353386 in page 20-21 in the subsection of “Ablation study” and line 762-780 in page 40 in the subsection of “Further potential redundancies removal between the training and the test”). The result shows that even when using “10% sequence identity” as the threshold, mean precisions of the predicted contacts only decreases by ~3% (Table 2).

(2) Figures with head-to-head comparison scatter plots are hard to understand as scatter plots because too many different methods are abstracted into a single plot with multiple colors. It would be better to provide individual head-tohead scatter plots as supplementary figures, not in the main figure.

We thank the reviewer for the suggestion! We will include the individual head-to-head scatter plots as supplementary figures in the revision (Figure S1 and Figure S2 in the supplementary).

(3) The authors claim that PLMGraph-Inter is complementary to AlphaFoldmultimer as it shows better precision for the cases where AlphaFold-multimer fails. To strengthen the point, the qualities of predicted complex structures via protein-protein docking with predicted contacts as restraints should have been compared to those of AlphaFold-multimer structures.

We thank the reviewer for the suggestion! We included this comparison in the revision(Figure S7).

(4) It would be interesting to further analyze whether there is a difference in prediction performance depending on the depth of multiple sequence alignment or the type of complex (antigen-antibody, enzyme-substrates, single species PPI, multiple species PPI, etc).

We thank the reviewer for the suggestion! We analyzed the relationship between the prediction performance and the depth of MSA in the revision (Figure S4 and Line 253264 in page 15 in the subsection of “Evaluation of PLMGraph-Inter on HomoPDB and HeteroPDB test sets” and line 798-806 in page 42 in the subsection of “Calculating the normalized number of the effective sequences of paired MSA”).

**Reviewer #2 (Recommendations For The Authors):**
I have the following suggestions in addition to the public review.(1) Overall, the manuscript is well-written; however, I recommend a careful review for minor grammar corrections to polish the final text.

We carefully checked the manuscript and corrected all the grammar issues and typos we found in the revision.

(2) It would be better to indicate that single sequence embeddings, MSA embeddings, and structure embeddings are ESM-1b, ESM-MSA & PSSM, and ESM-IF when they are first mentioned in the manuscript e.g. single sequence embeddings from ESM-1b, MSA embeddings from ESM-MSA and PSSM, and structural embeddings from ESM-IF.

We revised the manuscript according to the reviewer’s suggestion (line 86-88 in page 6; line 99-101 in page 7).

(3) I don't think "outer concatenation" is commonly used. Please specify whether it's outer sum, outer product, or horizontal & vertical tiling followed by concatenation.

It is horizontal & vertical tiling followed by concatenation. We clarified this in the revision (line 129-130 in page 8).

(4) 10th sentence on the page where the Results section starts, please briefly mention what are the other 2D pairwise features.

We clarified this in the revision (line 131-132 in page 8).

(5) In the result section, it states edges are defined based on Ca distances, but in the method section, it says edges are determined based on heavy atom distances. Please correct one of them.

It should be Ca distances. We are sorry for the carelessness, and we corrected this in the revision (line 646 in page 35).

(6) For the sentence, "Where ESM-1b and ESM-MSA-1b are pretrained PLMs learned from large datasets of sequences and MSAs respectively without label supervision,", I'd suggest replacing "without label supervision" with "with masked language modeling tasks" for clarity.

We revised the manuscript according to the reviewer’s suggestion (line 150-151 in page9).

(7) It would be better to briefly explain what is the dimensional hybrid residual block when it first mentioned.

We explained the dimensional hybrid residue block when it first mentioned in the revision (line 107 in page 7).

(8) Please include error bars for the bar plots and standard deviations for the tables.

We thank the reviewer for the suggestion! Our understanding is the error bars and standard deviations are very informative for data which follow gaussian-like distributions, but our data (precisions of the predicted contacts) are obviously not this type. Most previous studies in protein contact prediction and inter-protein contact prediction also did not include these in their plots or tables. In our case, including these elements requires a dramatic change of the styles of our figures and tables, but we would like to not change our figures and tables too much in the revision.

(9) Please indicate whether the chain break is considered to generate attention map features from ESM-MSA-1b. If it's considered, please specify how.

The paired sequences were directly concatenated without using any letter to connect them, which means we did not consider chain break in generating the attention maps from ESM-MSA-1b.